# Automated highly multiplexed super-resolution imaging of protein nano-architecture in cells and tissues

Maja Klevanski [1], Frank Herrmannsdoerfer[1], Steffen Sass [1], Varun Venkataramani[1], Mike Heilemann [1,2] & Thomas Kuner [1✉]

Understanding the nano-architecture of protein machines in diverse subcellular compartments remains a challenge despite rapid progress in super-resolution microscopy. While single-molecule localization microscopy techniques allow the visualization and identification of cellular structures with near-molecular resolution, multiplex-labeling of tens of target proteins within the same sample has not yet been achieved routinely. However, single sample multiplexing is essential to detect patterns that threaten to get lost in multi-sample averaging. Here, we report maS$^3$TORM (multiplexed automated serial staining stochastic optical reconstruction microscopy), a microscopy approach capable of fully automated 3D direct STORM (dSTORM) imaging and solution exchange employing a re-staining protocol to achieve highly multiplexed protein localization within individual biological samples. We demonstrate 3D super-resolution images of 15 targets in single cultured cells and 16 targets in individual neuronal tissue samples with <10 nm localization precision, allowing us to define distinct nano-architectural features of protein distribution within the presynaptic nerve terminal.

[1] Department of Functional Neuroanatomy, Institute for Anatomy and Cell Biology, Heidelberg University, Im Neuenheimer Feld 307, 69120 Heidelberg, Germany. [2] Institute of Physical and Theoretical Chemistry, Goethe-University Frankfurt, Max-von-Laue-Str. 7, 60438 Frankfurt, Germany. ✉email: thomas.kuner@uni-heidelberg.de

Proteins in cells interact as networks in space and time. Knowing the molecular organization of proteins relative to each other is crucial to understand mechanisms of cellular function. This advance requires (i) imaging techniques that are capable to visualize cellular structures with near-molecular resolution, (ii) strategies for multi-protein staining of an individual biological sample, and (iii) an automated workflow for non-invasive solution exchange and data acquisition. Identifying complex nano-architectural patterns of 3D protein arrangement requires the localization of as many as possible different proteins within a single sample because averaging distribution patterns determined from many individual samples may mask the underlying organizational principle, at least for structures that are not inherently symmetrical.

Optical super-resolution microscopy has revolutionized cell biology (reviewed in Sigal et al.[1]). One branch of techniques is single-molecule localization microscopy (SMLM)[2], comprising (direct) stochastic optical reconstruction microscopy ((d)STORM)[3,4], photoactivated localization microscopy (PALM)[5], points accumulation for imaging in nanoscale topography (PAINT)[6], and DNA-PAINT[7]. SMLM achieves a localization precision of <10 nm, allowing to decipher protein nano-architecture in 3D, and intrinsically providing access to molecule numbers[8].

Multiplexed protein labeling remains a challenge to all fluorescence microscopy methods. Stationary labels are restricted owing to spectral limitations, on the one side, and lack of efficient elution protocols, on the other. Only few multi-color SMLM reports exist so far[9–11]. However, none of these approaches allowed for automated multiplexing in cells; multiplexed SMLM in tissue was so far not reported. Although Yi et al. (2016) were able to image 25 epitopes in cells, their approach is restricted to labeled primary antibodies that are time-consuming to produce and not possible for all antibodies. Dynamic labels, such as reversibly binding fluorophore-labeled DNA strands as used in DNA-PAINT, are ideally suited for multiplexed imaging[12,13]. However, multiplexing with DNA-PAINT is also restricted to available primary labels, such as antibodies or aptamers[14]. Multiplex imaging at large scale inevitably requires automated solution exchange, sample positioning, image acquisition, and re-staining techniques. For automation, Almada[15] employed the NanoJ fluidics system to perform a five-color experiment. Another recent approach successfully demonstrated high-throughput STORM screening of cells in multiwell plates[16]. However, this approach does not allow multiplex analysis within individual samples. To date, a fully automated microscopy system including non-invasive solution exchange is not available.

The nano-architecture of active zones (AZs), presynaptic specializations mediating neurotransmitter release, has been addressed in studies mostly focusing on the molecular architecture at the *Drosophila* neuromuscular junction[17]. Mammalian central synapses and their nano-architecture have been studied in synaptosomes[18] and cultured hippocampal neurons[19–22]. However, only few studies addressed presynaptic protein topography in mature synapses of nervous tissue, such as the calyx of Held[23], using super-resolution microscopy[24–26]. A systematic investigation of molecular nano-architecture in natively matured synapses has not yet been achieved. SMLM imaging of neuronal tissue[25–29] and even more multiplexing remain challenging because of technical limitations.

Here, we developed a universal method for serial detection of multiple targets that is suitable for cells and tissues. Our fully automated system comprises of an in-house built 3D *d*STORM setup, a pipetting robot, and a custom-written software allowing coordinated operation of both systems. We illustrate that maS[3]TORM can execute re-staining protocols to perform multiplex experiments with at least 15 targets that could be detected in single cells and 16 targets in neuronal tissue. Moreover, we developed and integrated tools for multi-protein spatial analysis, allowing us to study the protein landscape within individual synapses. For example, the localization of motor protein myosin Va (MyoVa) is correlated with synaptic vesicle (SV) marker VGlut1 throughout the synapse, suggesting that in addition to its role in SV trafficking, MyoVa could have a role in SV clustering and tethering. Furthermore, we observe that poly-merized actin is less abundant in close proximity to membranes with AZs compared with AZ-free membrane stretches, in line with observations that actin dynamics regulates neuronal activity[22,30–32].

## Results

**maS[3]TORM setup and multiplexing workflow.** maS[3]TORM is composed of a home-built *d*STORM setup and the commercially available pipetting robot PAL3 RTC (Fig. 1a, b; Supplementary Figs. 1, 2 and Supplementary Movie 1). The setup is designed for astigmatism-based 3D measurements[33] and allows simultaneous acquisition of two channels by spectral demixing (Supplementary Fig. 1)[34–36]. For axial drift correction, it is equipped with a focus stabilization module (Fig. 1a, b, Supplementary Fig. 1). To ensure an automated and coordinated interplay between the staining procedures carried out by the pipetting robot and image acquisition, the Experiment Editor software was developed that can address both the μManager[37] based Microscope Control software and the commercial PAL3 Chronos software (Supplementary Figs. 1–4). In the Experiment Editor, the user can create customized experimental workflows by combining microscope- and robot-related modules and iterating tasks for the desired number of times (Supplementary Figs. 3, 4).

For multiplex experiments, we developed re-staining workflows allowing fixed cells or tissues to be serially labeled, imaged, and stripped (Fig. 1c). Once the exact experimental workflow is designed and programmed using the Experiment Editor, the designated vials of the PAL3-pipetting robot can be filled with respective solutions and the fixed sample can be placed onto the motorized stage. After the initial staining is performed via Experiment Editor, regions of interest (ROIs) are selected manually and their coordinates are entered into the predesigned workflow in Experiment Editor using a specialized ROI module. To make sure that the initially selected focal plane can be retrieved in each imaging round, we developed an auto-refocusing system (for details, see Methods) that is based on imaging of a reference region with fluorescent TetraSpeck beads. Next, dual-color 3D *d*STORM data of the preselected ROIs are acquired and automatically redirected to and processed by rapidSTORM[38]. For precise registration of images from distinct staining rounds, we imaged fiducial beads that were seeded underneath cells or tissue for maximal immobilization. Fluorescence of the fiducials could be preserved for at least 11 imaging and bleaching rounds by moderately exciting them using the 561 nm laser while recording through the 700/75 bandpass filter (same filter as used for *d*STORM acquisition) for minimal chromatic aberration. Feature-based lateral drift correction within individual super-resolution images was implemented in the subsequent post-processing workflow.

Depending on the type of labels used, signal removal was achieved by photo-bleaching with 405 nm and 661 nm laser light[10] or both photo-bleaching and chemical elution (for details, see Methods). The parameter space for signal removal was probed systematically, defining a bleaching time of 4 min and a duration of 15 min for elution using a 0.1% sodium dodecyl sulphate (SDS) buffer at pH 13 (modified from Collman et al.[39]) as the most

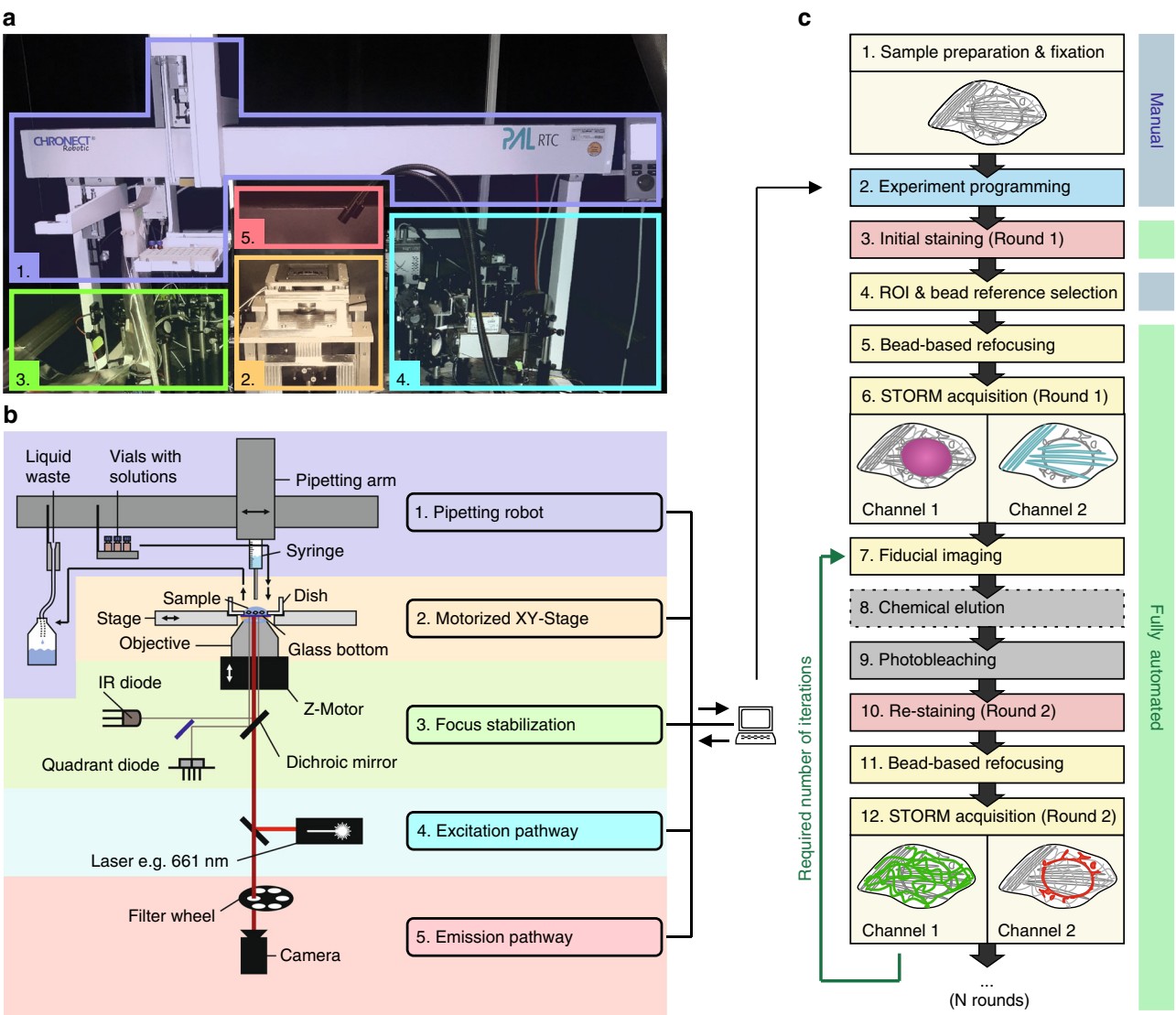

**Fig. 1 Automated maS³TORM multiplex setup and experimental workflow.** Photographic **a** and schematic overview **b** of the setup body comprising the pipetting robot (1, purple), the motorized stage on top of the inverted 1.49 NA 100x objective (2, yellow), the focus stabilization system (3, green), components of the excitation (4, blue) and the emission pathway (5, red). **c** Experimental workflow for an automated multiplex approach. From step 5 onwards, the workflow is fully automated. Step 8 is conditional (stippled line). Details on the series of steps in an automated imaging session see Methods.

suitable condition for multiplexing experiments (Supplementary Fig. 5). Control experiments with U2OS cells labeled with Tom20 antibody showed that bleaching successfully removed 98.7 ± 1% of the initial signal (Supplementary Figs. 6 and 7, Supplementary Note 1). However, if primary antibodies of the same species are used in consecutive staining rounds, bleaching is not sufficient. In this case, antibodies need to be removed physically by elution in addition to bleaching the fluorophores. After elution and bleaching, only 1.9 ± 1% of the initial Tom20 signal can be retrieved by re-application of the secondary antibody in the presence of a competing primary antibody (Supplementary Fig. 6). Variable propensity for cross-talk of different antibodies and labels (Supplementary Fig. 8) emphasizes the importance of customized strategies for high-fidelity multiplexed imaging. Therefore, we provide some general rules to guide the user to a successful experimental design (see Supplementary Note 2). Notwithstanding, each antibody used for a multiplexing experiment needs to be characterized in detail beforehand. Next, we verified the re-staining efficiency by labeling, eluting, and imaging mitochondria in U2OS cells and tissue, using the same antibody

in 10 subsequent rounds. Regression analysis revealed a loss of 2.7% and 4.1% per imaging round for cells and tissue, respectively (Supplementary Fig. 9a–d). Finally, we show that the cross-correlation of images obtained from 10 consecutive rounds relative to the first round remained unchanged, suggesting that the overall structural integrity of the sample was well maintained (Supplementary Fig. 9e–g). In summary, these results demonstrate the high fidelity of the re-staining strategy devised here.

**Multiplexed 3D super-resolution imaging of single cells**. We next demonstrate multiplexed 3D *d*STORM imaging in fixed U2OS cells. Within 56 h, we performed 11 rounds of staining, and extracted 15 high-quality super-resolution data sets (Fig. 2a; see Supplementary Table 1 for a detailed workflow). The image quality was assessed using two different metrics, localization precision and spatial resolution. The localization precision of SMLM images shown in Fig. 2a remained stable at a high grade with a mean value of 7.5 ± 2 nm (calculated according to the nearest neighbor distance method[40]; Supplementary Fig. 10a). The spatial resolution was determined by decorrelation analysis[41]

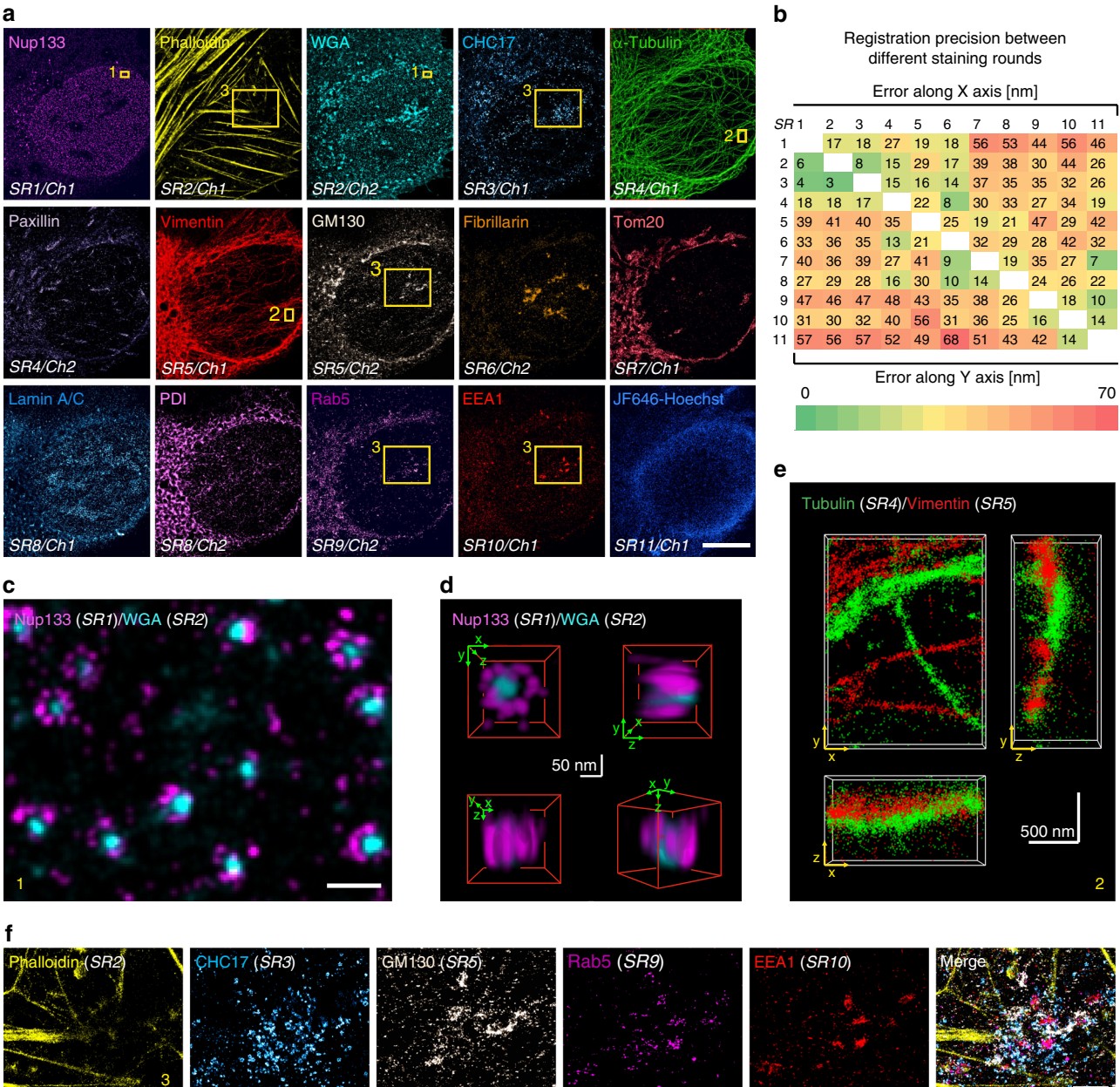

**Fig. 2 Multi-target 3D dSTORM of a single U2OS cell. a** A U2OS cell with 15 targets labeled and acquired by 11 staining rounds (SR). Two channels (Ch) can be detected per SR, but not always each channel can be used for final analysis. In addition, to get optimal signal for some sensitive labels (e.g., GFP nanobodies against the Ypet-tag of Nup133), it was beneficial to not stain targets in the second channel (see Supplementary Note 2). Boxes denote regions magnified in **c**, **e**, and **f**. Another three experiments were carried out with similar designs and outcomes. Further details see Methods section. **b** Heatmap representing the registration error along the $x$ (upper matrix part) and the $y$ axis (lower matrix part) reveals precise registration, in particular, between images of adjacent staining rounds (arranged along the white diagonal fields). **c** Magnified view of boxed region 1 in **a** with merged peripheral Nup133 and central WGA nuclear pore labeling demonstrating that super-resolution images from consecutive SRs can be registered with high precision. **d** 3D views of a nuclear pore complex (from the multiplex experiment in **a**) demonstrate the high lateral and axial resolution and registration precision. **e** Merged tubulin and vimentin 3D images from consecutive SRs, demonstrating that super-resolution images can be acquired and registered in 3D (boxed region 2 in **a**). **f** Five-color merge of boxed region 3 in **a**: phalloidin-labeled actin filaments, CHC17-labeled clathrin-coated structures, GM130-labeled proximal Golgi apparatus, endosome-associated proteins Rab5 and EEA1. Scale bars correspond to 10 μm in **a**, 200 nm in **c**, 50 nm in **d**, 500 nm in **e**, and 2 μm in **f**.

and largely ranged between 25 and 30 nm (Supplementary Fig. 10b, c). Accuracy of structural resolution and metrics was further confirmed by quantifying the diameter of nuclear pore complexes (Supplementary Fig. 10d). The targets for the multiplex experiment comprised Nup133 of the nuclear pore complex, F-actin, sialic acid, and $N$-acetylglucosaminyl residues (bound by wheat germ agglutinin (WGA)) among others present at the cell membrane and in the center of nuclear pores, CHC17 of clathrin-

coated vesicles, α-tubulin, paxillin in focal adhesions, intermediate filament vimentin, GM130 in Golgi, fibrillarin in nucleoli, Tom20 in mitochondria, lamin A/C at the nucleus, PDI in the ER, Rab5, and EEA1 in early endosomes, and DNA labeled by $JF_{646}$-Hoechst conjugate[42]. All images could be successfully registered to each other with good precision (Fig. 2b). The quality of the images and the registration process are illustrated in Fig. 2c, d showing Nup133 signal localized at the peripheral

subunits of nuclear pores merged with WGA signal in the center of nuclear pores. The 3D views reveal that the Nup133 signal originates from two layers/rings of the nuclear pore complex (lower right panel in Fig. 2d). All data were collected in 3D mode and can be registered along all three axes as exemplified by α-tubulin and vimentin in Fig. 2e (for 2D renderings, see Supplementary Fig. 10) that partially run parallel to each other but do not colocalize (for other 3D examples, see Supplementary Fig. 11 and Supplementary Movie 2). Finally, magnified dSTORM images of endosome-related and -adjacent structures (Fig. 2f) prove that early endosomes can be successfully re-stained by two different endosomal markers, Rab5 and EEA1. As expected, markers for Golgi, clathrin-coated structures, and associated actin patches[43,44] can be found in close proximity. All the above examples demonstrate a robust, multi-target super-resolution imaging using maS³TORM.

**Multiplexed imaging of 16 targets in neuronal tissue.** We next applied multiplex imaging to brain tissue containing a well-established glutamatergic model synapse referred to as the calyx of Held[23]. The giant presynaptic calyx of Held envelopes the postsynaptic principal cell (Fig. 3a) and forms swellings surrounding the oval-shaped postsynaptic cell in a ring-like arrangement (cross-section in middle panel) with AZs situated at the synaptic calyx border (lower panel). Figure 3b illustrates the arrangement of calyces imaged in Fig. 3c. Tissue sections with a thickness of 400 nm were obtained using a cryo-ultramicrotome and allowed for direct access of antibodies to the target sites without the use of detergents (see Methods). In 10 consecutive staining rounds we were able to super-resolve the distribution of the following 16 targets in situ (Fig. 3c): AZ marker bassoon, postsynaptic density marker homer1/2/3, F-actin, sialic acid, and N-acetylglucosaminyl residues among others present at the cell membrane and in Golgi, CHC17 in clathrin-coated structures, MAP2 at the postsynapse, SV marker VGlut1, GM130 in Golgi, presynaptic Rab3a, Tom20 in mitochondria, α-tubulin, γ-actin, SV marker synaptophysin, α/β-synuclein, PDI in ER, and DNA labeled by JF$_{646}$-Hoechst (see Supplementary Table 2 for a detailed workflow and Supplementary Table 3 for list of used antibodies and labels). Individual dSTORM images could be registered with excellent precision (Fig. 3d) and exhibited a spatial resolution ranging from 25 to 30 nm (Supplementary Fig. 10c). These data demonstrate the feasibility of imaging the distribution of 16 individual protein targets within a single tissue volume at super-resolution.

**maS³TORM uncovers global and local protein sociology.** For quantitative analyses of selected cytoskeletal and synaptic proteins at the calyx of Held, we conducted five independent multiplex experiments. As our main reference for synaptic geometry we used the WGA marker that proved to efficiently label the calyceal borders. Close-up dSTORM images in Fig. 3e demonstrate that WGA signal at the membrane facing the postsynapse is sandwiched between presynaptic bassoon and postsynaptic homer. As expected, SV marker VGlut1 fills the presynaptic space indicated by WGA. Exemplary images suggest that actin filaments show a rather peripheral localization while microtubules can be found further inside the calyx. These observations are also reflected in the averaged line profiles (Fig. 3f) laid across the calyx lumen perpendicularly to the presynaptic membrane. Average plots were obtained from a total of 15 calyces imaged in five independent experiments. For statistical reliability, 4–16 line profiles per calyx were analyzed. Calyceal borders were determined based on the WGA signal and all data were normalized according to the mean calyx thickness of 987.2 ± 238.6 nm (with

0 nm corresponding to the inner synaptic and 987.2 nm to the outer border; for details on line profile analysis, see Methods and Supplementary Fig. 12). Interestingly, βII-spectrin shows a similar spatial distribution as F-actin, and MyoVa resembles the spatial distribution of VGlut1. Quantitative analysis of the spatial patterns of α-tubulin, F-actin (two peaks: p1 and p2), and VGlut1 curves confirms a significant difference of the peak locations (Fig. 3g). Compared with the synaptic vesicle signal visualized by VGlut1, which on average is homogeneously distributed throughout the presynaptic space, the peak α-tubulin (α-Tub) signal is located in the center of the calyx interior with a slight tendency towards the outer synapse membrane. At least two peaks can be discriminated for F-actin: in close proximity to the outer and the inner presynaptic membrane. Part of the contribution to the second peak comes from the membrane-proximal part of the postsynaptic cell. Interestingly, βII-spectrin follows a similar distribution as F-actin and MyoVa is distributed similarly to VGlut1 (for line profiles of all proteins analyzed, see Supplementary Fig. 13a–d).

Next, we investigated if protein distributions differed between line profiles drawn through AZ-containing calyx parts vs. AZ-free areas, each reflecting areas of specialized function (see Supplementary Fig. 12b). Indeed, we encountered more VGlut1 expression within a distance of up to 500 nm from the AZs (Fig. 3h), consistent with the distribution of SVs at sites of exocytosis known from electron microscopy[45]. Interestingly, less polymerized actin was found in close proximity to membranes containing AZs as compared with AZ-free membrane stretches (Fig. 3i). Similar to VGlut1, MyoVa was enriched at AZ-proximal regions, consistent with its role in SV trafficking (Fig. 3j; see Supplementary Fig. 13e–o for complete data set for AZ-specific distribution). Exemplary plots of two individual AZs (Fig. 3k, l) demonstrate that different AZs can have differing fingerprints and further confirm a correlation between VGlut1 and MyoVa.

Finally, to gain an overview of the global protein interrelationship, we conducted a large-scale colocalization analysis by determining Pearson's r coefficients for all possible protein pairs at AZ-proximal vs. AZ-free calyx patches (Fig. 3m; Supplementary Fig. 14). Although synaptophysin and VGlut1 or different tubulin variants show expected colocalization with each other, irrespective of whether they are proximal to the AZ or not, other proteins differed significantly in their colocalization coefficient depending on their relation to the AZ. In agreement with the previous notion that less polymerized actin was found in AZ proximity, F-actin showed significantly higher colocalization with WGA in AZ-free calyx regions. These results are consistent with previous work, suggesting that actin dynamics is linked to synaptic release[22,31,32,46], yet, to our knowledge, so far this arrangement has not been visualized directly. Also the correlation between MyoVa and VGlut1 already observed in the line profile analysis (Fig. 3f) can be confirmed by high colocalization coefficients, but is not significantly dependent on the proximity to AZs.

Based on our analyses of the overall (Fig. 3f, g, Supplementary Fig. 13a–d) and the AZ-dependent distribution (Fig. 3h–m, Supplementary Fig. 13e–o) of multiple targets at the calyx synapse, we propose a model (Fig. 4), wherein F-actin and βII-spectrin are predominantly found at the calyceal borders with less F-actin at SV release sites. Tubulin density in AZ-containing calyx sections is higher in the center of the calyx interior. SVs along with MyoVa are distributed throughout the calyx but show both increased levels in AZ proximity, suggesting that MyoVa does not only play a role in the initial transfer of SVs from microtubules to F-actin, but also in the later steps of the synaptic vesicle cycle, possibly including clustering and exocytosis. Although this model as a whole may appear familiar, it is the first of its kind that is derived from a set of 14 proteins identified

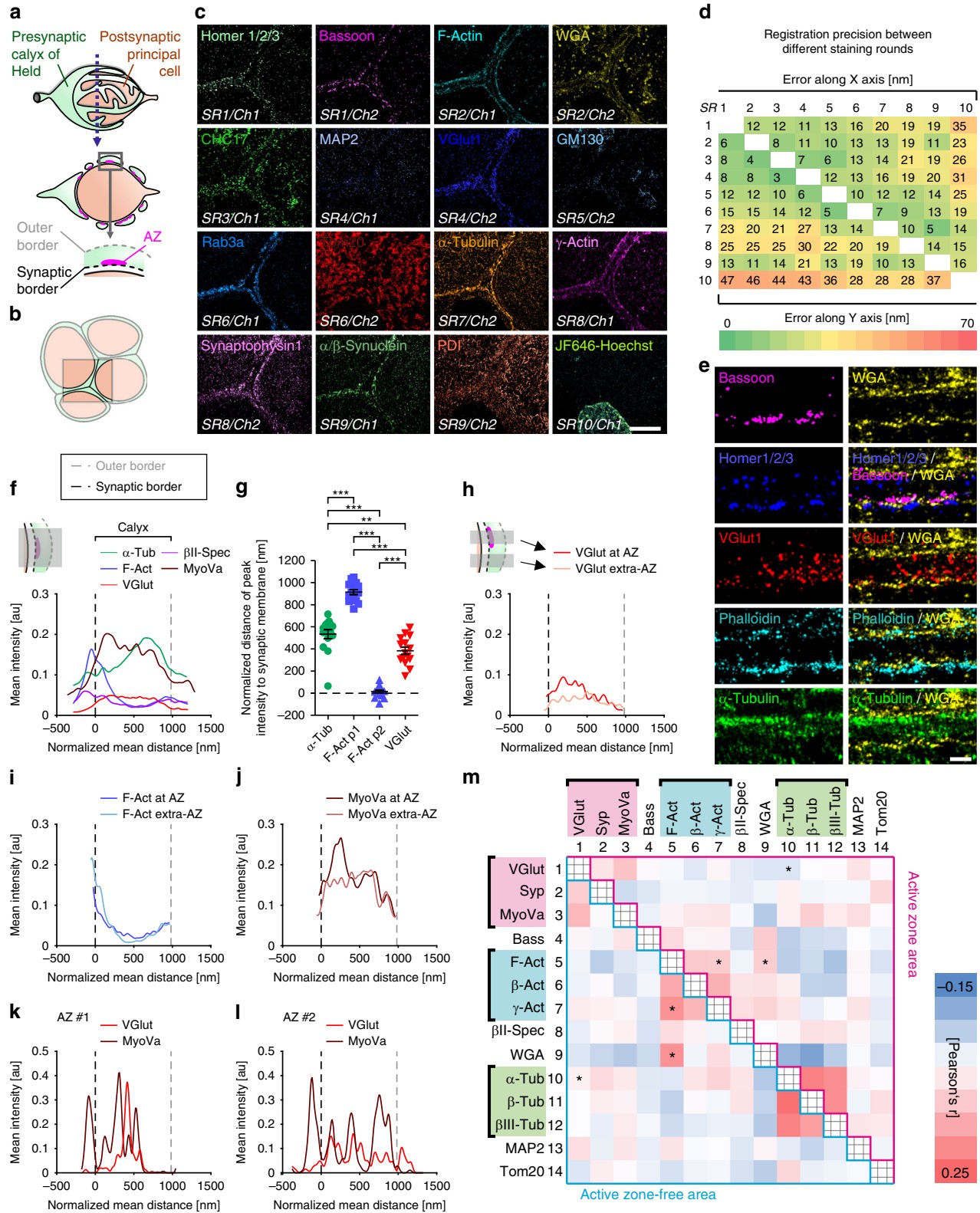

at tens of nanometer resolution in the same physical sample, rather than reflecting a population average obtained from literally hundreds of different samples.

## Discussion

The maS³TORM strategy presented here successfully combines the advantages of three powerful approaches: super-resolution microscopy, multiplexing by re-staining protocols, and automation. Careful consideration of label properties and thereof arising order of label application and removal techniques (see Supplementary Note 2) allowed us to detect numerous targets in individual samples. We were able to automatically conduct high-content multiplexed SMLM imaging experiments in cells and neuronal tissue. Using our approach, we gained important

**Fig. 3 Multiplex dSTORM of the calyx of Held synapse. a** Schematic representation of the calyx of Held synapse with the cross-section revealing multiple swellings with active zones (AZ) and zoom-in delineating borders relevant for analysis. **b** Cartoon illustrating the topology of adjacent calyces as represented in **c**. **c** STORM images showing 16 targets of the same section acquired in 10 consecutive staining rounds (SR) using one or two spectral channels (Ch). **d** High registration precision (3–47 nm) of images shown in **b** achieved throughout all SRs. **e** Magnified images of cross-section of calyces (as depicted in **a**). Homer, bassoon, and WGA stainings confirm imaging plane-perpendicular orientation. **f** Presynaptic distribution of selected proteins analyzed and averaged over 9–15 calyces from 3 to 5 experiments. Calyx borders derived from WGA staining. **g** Distances of α-Tub peak, first and second F-Act peak (p1, p2), and VGlut1 peak to synaptic membrane differed significantly (**$p \leq 0.01$; ***$p \leq 0.001$, one-way ANOVA). **h–j** Differential protein distribution within regions containing or lacking AZs (see inset in **h**). **h** Synapse-proximal regions contain more synaptic vesicles (SVs) (VGlut1 staining) at AZ sites as compared with extra-AZ regions. **i** Less polymerized actin in AZ-proximal regions. **j** More MyoVa at AZ sites, correlating with VGlut1 profile. **k, l** Example of single AZs supports correlation between SV and MyoVa distribution and illustrates individuality of AZs. **m** Colocalization matrix showing Pearson's $r$ coefficients (exact values, see Supplementary Fig. 14) between all combinations of 14 analyzed proteins at AZ-proximal (upper matrix part) and AZ-free (lower part) calyx regions. Asterisks denote statistical significance for colocalization of selected protein pairs (*$p \leq 0.05$; unpaired two-tailed Student's $t$ test). Scale bars correspond to 5 µm in **c** and 500 nm in **e**. Scatter plot **g** shows mean ± SD. Note that in addition to representative images in **c** and **e** another six and three experiments were carried out, respectively, with similar designs and outcomes. For exact sample numbers, see Supplementary Table 5. Source data, including exact $p$ values for **g** and **m**, are provided as a Source Data file.

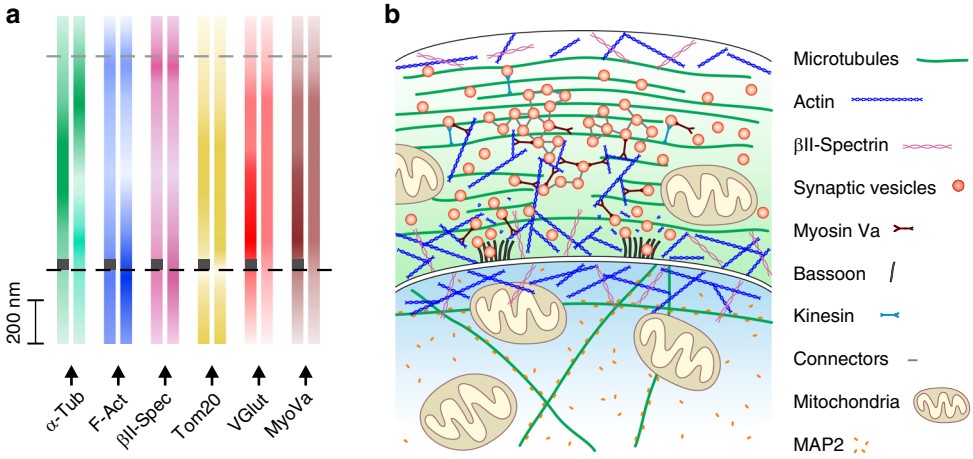

**Fig. 4 Model of protein distribution in the calyx of Held synapse. a** Distribution of proteins examined in Fig. 3 and Supplementary Fig. 13 indicated by color gradient. Distribution at active zone (AZ)-positive regions are indicated by dark gray rectangles at the bottom of color gradient bars. **b** Model of synaptic architecture at the calyx of Held. Predominantly peripheral distribution of F-actin paralleled by βII-spectrin distribution. Note the decreased presence of polymerized actin in AZ regions. Microtubules are distributed throughout the presynapse with AZ-containing regions more densely populated by tubulins. Synaptic vesicles (SVs) cover up the whole presynaptic space. At the presynaptic membrane, SVs are more abundant in AZ-proximal regions. The distribution of myosin Va (MyoVa) is highly correlated with the vesicular VGlut1 signal underpinning its role in SV trafficking: MyoVa is known to mediate the vesicular transfer from microtubules along which they are transported by kinesin motors to actin filaments. Interestingly, together with actin, MyoVa might be involved in SV clustering, similarly to connectors (structures of unknown molecular identity interconnecting SVs). In addition to the membrane-binding WGA lectin, the anatomy of the calyx synapse was traced using bassoon as the AZ marker. The mitochondrial protein Tom20 is present in pre- and postsynaptic compartments, whereas MAP2 is specifically expressed at the postsynapse.

insights on the multi-protein arrangement within single pre-synaptic terminals.

In contrast to spectral multiplexing, the maS³TORM approach avoids chromatic aberration by repeated usage of two fluorophores and spectral demixing. At the same time, we could drastically reduce the number of samples, which would otherwise be necessary to study protein interrelationships using the limited channel capacity of spectral multiplexing, and thus acquisition time. For example, being able to image only two targets at once, a researcher who wants to study 16 proteins, would have to conduct at least 120 single experiments without having the possibility to relate them to marker proteins such as WGA and bassoon required to clarify synaptic geometry. Furthermore, to conduct a comparable experiment and process the emerging data, researchers would need substantially longer working hours. By contrast, our approach minimizes the need of manual sample handling and ensures standardized conditions by automated liquid exchange. In addition, the looping functionality and modularity of the Experiment Editor enables an effective

realization of any number of iterations. The feasible number of staining rounds is restricted mainly by the availability of labels with high specificity that allow efficient removal, and in case of labeling by primary and secondary antibodies by the limited number of antibody host species. Thus, broad availability of fluorophores directly conjugated to primary antibodies or other labeling agents will not only reduce the overall label size and thereby increase the spatial resolution, but also greatly improve multiplexing capacity. DNA-PAINT is another strategy that can easily be implemented in our automated microscope and allows for a high degree of multiplexing[7].

Deriving molecular organizational nano-architecture of sub-cellular compartments requires determining the position of as many proteins as possible within one and the same sample because some architectural features will only become evident when looking at more than two to three proteins and because averaging of pair-wise localization images across multiple samples may obstruct patterns that lack evident symmetries. An important prerequisite to achieve single probe multiplexing rests in the

precision at which serial staining rounds can be superimposed, so that all proteins determined are placed in the same reference space. We demonstrate that in tissue this is possible with a precision ranging from 3 to 47 nm (Fig. 3d; see Supplementary Note 3 for a detailed discussion), a number that could be further improved by using spatially invariant fiducial markers. Thus, maS³TORM opens an avenue to systematically investigate complex nano-architectural principles of protein arrangement in single samples.

Applying maS³TORM to neuronal tissue, we were able to gain important insights into the nano-architecture of a glutamatergic synapse. To achieve this, we established a tissue preparation method for super-resolution imaging: 400 nm thick tissue sections that are large enough to contain complete structures of interest while providing direct access of antibodies to the cytosolic compartment without the use of detergents. This approach ensures excellent structural preservation of the tissue and provides ideal conditions for super-resolution imaging. In such thin tissue sections, we observed more polymerized actin at AZ-free membranes as compared with AZ-proximal regions. This is in line with functional studies showing that actin dynamics regulates neuronal activity[22,30–32]. βII-spectrin shows a spatial distribution similar to actin at the calyx of Held and is decreased at presynaptic specializations, suggesting that actin and βII-spectrin may cooperate in synapse-modeling functions. Furthermore, we found that myosin Va distribution is correlated with VGlut1 throughout the presynapse. In addition to its role in transferring SVs from microtubules to actin filaments[47], this may imply additional functions of MyoVa: (1) interconnecting of SVs and/or (2) SV tethering to the membrane. Such connections could be realized via MyoVa–actin interactions[32]. The electron microscopic study by Cole et al.[48] indicates that in addition to short SV connectors and tethers, longer cluster filaments are found either extending from the AZ or throughout SV clusters.

In conclusion, the maS³TORM approach proved to be an effective and universal method for studying highly complex biomolecular systems. Automated super-resolution multiplex solutions open promising avenues to consistently and systematically probe nano-architecture on a large scale in different fields of biology. With further development of labels, it will be possible to monitor more targets with even higher precision. Our data illustrate that elaborate computational methods employing artificial intelligence will be needed to infer organizational principles of protein nano-architectures from the distribution patterns of tens of proteins within an individual biological sample.

## Methods

**maS³TORM setup.** The custom-built 3D *d*STORM setup (Supplementary Fig. 1, parts referenced here as #x and summarized in Supplementary Table 4) is equipped with 405 nm (#1; Cube, Coherent), 488 nm (#2; Obis, Coherent), 561 nm (#3; Obis, Coherent), and 661 nm (#4; Cube, Coherent) lasers. For excitation, lasers were focused onto the back focal plane of a ×100 total internal reflection fluorescence objective with a numerical aperture of 1.49 (#9; Olympus). Emission light is collected by a multi-bandpass filter (#8; ZT405/488/561/647rpc, Chroma) and passes a cylindrical lens (#10; Thorlabs) that is used for astigmatism-based 3D imaging. Dual-color *d*STORM imaging was performed by illumination with the 661 nm laser. Light emitted from the sample was filtered through the 700/75 ET bandpass filter, split into two beams by a custom-built dichroic filter (#13; 690 nm, Chroma), and focused onto the EMCCD camera (#15; iXon Ultra 897, Andor) with each channel occupying one half of the camera chip. The *z*-focus was stabilized by a feedback loop. The beam of an infrared diode (#16; 785 nm, Thorlabs) is reflected at the edge of the cover glass surface and the sample medium and projected onto a quadrant photodiode (#18; Laser Components). Depending on the position at which the beam illuminates the quadrant diode, a respective signal is sent to a motorized objective positioner (Physik Instrumente) to move the objective along the *z* axis, thereby correcting for changes of the focal plane. To move the sample laterally, we used a custom-built motorized *xy* stage (SmarAct). To enable an automated on-microscope staining and elution procedure, we complemented the microscope setup by the pipetting robot PAL3 RTC (Axel Semrau). Our robot is equipped with three vial trays and two syringes (100 µl and 1000 µl) and allows pre-mixing of solutions and their application onto the sample as well as their precise and reliable removal from the glass bottom dish.

For fully automated microscopy and re-staining procedure, both microscope hardware elements as well as the pipetting robot were integrated into one custom-written control software, called Experiment Editor that can be conveniently addressed by the user via the corresponding graphical user interface (GUI). To this end, all microscope components were implemented into the device manager of the open source software µManager (Open Imaging)[37]. To conveniently use functionality of the hardware, initially, on top of µManager, the custom-written Microscope Control software was developed. The pipetting robot, in turn, can be controlled by the commercially available Chronos software (Axel Semrau). Although the Microscope Control software can be directly addressed by the Experiment Editor, the pipetting robot is controlled and synchronized indirectly by employing a custom-written plugin that enabled Chronos to obtain tasks generated by the Experiment Editor in an exchange folder (Supplementary Fig. 4a).

**Cell preparation.** U2OS human bone osteosarcoma epithelial cells constitutively expressing Nup133-Ypet (Ypet is a YFP variant detectable by GFP nanobodies[49]) were cultured in DMEM/F-12 (Gibco) supplemented with 10% FCS (Gibco) and 2% GlutaMAX (Gibco) at 37 °C in 5% $CO_2$. Cells were transferred to glass bottom dishes (P35G-0-14-C, MatTek) pretreated with 50 µg/ml fibronectin (Sigma) and 1:500 Tetraspeck fluorescent beads (T7279, Invitrogen). After 24 h cells were washed twice with PSB and chemically fixed with 4% paraformaldehyde/4% sucrose dissolved in phosphate-buffered saline (PBS) for 15 min. After three washing steps with PBS, cells were permeabilized using 0.1% Triton in PBS for 10 min at room temperature (RT).

**Tissue preparation.** All experiments on animals were conducted in compliance with the German animal welfare guidelines according to protocol G-75/15, and approved by the Regierungspraesidium Karlsruhe. Young (12 days old) Sprague-Dawley rats (Charles River) were anesthetized and transcardially perfused using ~20 ml PBS, followed by ~30 ml 4% paraformaldehyde/PBS. Brains were removed and post-fixed in 4% paraformaldehyde/PBS for 24 h at 4 °C. The brain stem was cut into 200 µm thick slices using a vibratome (SLICER HR2, Sigmann-Elektronik). Using a scalpel, the medial nucleus of the trapezoid body (MNTB) was cut out, resulting in a ~(200 µm)³ tissue block that was infiltrated in 2.1 M sucrose in 0.1 M cacodylic acid buffer (pH 7.4) for 1 h at RT. The tissue was placed on a specimen holder, plunge-frozen in liquid nitrogen, and further sliced into 400 nm sections using a cryo-ultramicrotome (Ultracut S with cryo-chamber EM FCS, Leica). Thin sections were picked up using a 2.3 M sucrose (in cacodylic acid buffer) droplet in a metal loop and transferred to glass bottom dishes that were previously incubated with 1:500 Tetraspeck beads. Prior to staining, tissue was thawed for 10 min at RT and sucrose was removed by three washing steps with PBS for 15 min. This method of tissue preparation ensures good tissue preservation and direct access of antibodies to epitopes within the cell without the use of detergents[26]. Accessibility to the intracellular space is established by cutting the tissue into thin sections of 400 nm thickness, thereby opening all cells and even their sub-compartments such as dendritic processes or presynaptic terminals.

**Series of steps in an automated imaging session.** Each experiment consists of a series of steps illustrated in Fig. 1c. After preparing and fixing the sample (1), the experimental design is programmed using the custom-written Experiment Editor (2), the sample is stained for the first time (staining round 1) either manually or semi-automatically using the pipetting robot (3), and vials with solutions for labeling, washing, and elution are prepared. Subsequently, ROIs are selected for acquisition (4). To prevent focus loss during a long-term experiment, an automated refocusing approach maintains the focal plane based on a preselected region containing fluorescent bead fiducials (5). Then, *d*STORM 3D images of two channels are acquired and spectrally demixed (6). The new round starts with the acquisition of bead fiducials for subsequent registration of images from different staining rounds (7). To remove labeling from the imaged staining round, depending on the type of labels used, chemical elution (8) and/or photo-bleaching (9) is performed. Thereafter, the sample is re-stained using a new set of labels. (10) Refocusing (11) precedes *d*STORM 3D image acquisition. Finally, the procedure can be repeated as many times as desired by resuming with step 7. After a completed multiplex session, data sets were individually evaluated regarding labeling and image quality, and high-quality SMLM images were qualified for further analysis.

**Experiment Editor.** All steps described below were carried out using the custom-written Experiment Editor software (Supplementary Fig. 2). Its GUI allows convenient and coordinated operation of microscope components and the pipetting robot (Supplementary Fig. 3). Microscope control was realized via custom-written Microscope Control software, which, in turn, addresses µManager, an open source software, where all microscope hardware components were integrated. The pipetting robot was operated using a handshake procedure communicating via a specific exchange folder (Supplementary Fig. 4). A plugin that was custom-written for the commercial Chronos software constantly checks the content of an exchange folder for a path to a Chronos-tailored task list generated by Experiment Editor.

The task list consists of a set of commands that is loaded into Chronos and executed by the robot. After the tasks are successfully completed, a text file with specific key words is saved to the exchange folder by the Chronos plugin. This file, in turn, instructs the Experiment Editor to continue with the execution of the next workflow module.

**Staining and re-staining procedure**. Cells and cryo-sections were both blocked in 5% fetal calf serum (FCS) for 10 min. For immunohistochemistry, antibodies were applied in 0.5% FCS for 1 h. After each application of antibodies or other labels, samples were washed three times with PBS for a total of 15 min. Alexa Fluor 647 conjugated nanobodies against GFP and CHC17 against clathrin heavy chain 1 were applied for 1 h in 0.5% FCS. Alexa Fluor 647 conjugated phalloidin and CF680 conjugated WGA were applied in PBS for 20 min. $JF_{646}$-Hoechst was directly added to the imaging buffer. For the precise multiplexed STORM workflow used for U2OS cells (Fig. 2a) and for MNTB slices (Fig. 3c), see Supplementary Tables 1 and 2, respectively. All antibodies and other staining reagents used for cells and neuronal tissue are summarized in Supplementary Table 3. In cells and neuronal tissue, re-staining was realized by photo-bleaching and/or chemical elution. Elution of antibodies was performed using 0.1% SDS at pH 13 for 15 min (modified from Collman et al.[39]). To ensure the desired concentration, the sample was once washed with elution buffer before it was applied for a second time for incubation. After elution, the sample was washed with PBS six times in total. Bleaching was realized in PBS using the 661 nm laser at 100% power (~2 kW cm$^{-2}$) and 405 nm laser at 50% power (0.7 kW cm$^{-1}$) for 4 min[10]. All staining, bleaching, and elution procedures were carried out by the pipetting robot at RT (21 °C).

**SMLM image acquisition**. For all imaging rounds, β-mercaptoethylamine (MEA) buffer at pH 8 containing 100 mM MEA and 15 mM KOH in PBS was freshly prepared by the pipetting robot. To prevent dilution of MEA buffer by residual liquid inside the glass bottom dish, before imaging, the sample was once washed with MEA buffer. Conventional wide field images were acquired using 661 nm laser at low power (~1 W cm$^{-2}$) with an exposure time of 200 ms. For dSTORM images, photoswitching of fluorophores was initiated by 10 s exposure to 661 nm laser at high power (~2 kW cm$^{-2}$), followed by the acquisition of 20,000 frames at an exposure time of 30 ms. PAINT imaging with $JF_{646}$-Hoechst was performed at an exposure time of 50 ms. All images were recorded in astigmatism-based 3D mode and automatically, via a Python script, redirected to and processed by rapidSTORM.

**Fiducial imaging**. For precise registration of images from distinct staining rounds, we imaged fluorescent bead fiducials (excitable at 560 nm, 660 nm, and other wavelengths; Tetraspeck beads T7279, Invitrogen). To ensure maximal immobilization, we seeded the fiducials onto glass bottom dishes underneath the cells or tissue. As STORM images are performed at very high 661 nm laser intensity, bead fluorophores with an excitation maximum of 660 nm are bleached after approximately two STORM and two bleaching rounds. To preserve fiducials for >10 imaging and bleaching rounds, we used weak excitation by the 561 nm laser. Recording through a filter (700/75 ET Bandpass, AHF: F47-700) optimized for the 661 nm laser allowed us to minimize chromatic aberration. ROIs for SMLM imaging were chosen such that they contained at least five fiducials to allow precise registration.

**Auto-refocusing system**. In addition to the infrared diode-based focus stabilization system (described in the first paragraph of Methods) that ensures a constant basic position of the objective relative to upper edge of the cover glass, an automated focus recovery was implemented based on Tetraspeck fiducials. A reference region with at least five fluorescent beads in focus (without cells or tissue) was selected and its coordinates were transferred to the automated refocusing module. Subsequently, the major multiplex workflow was launched. In our design applied for cells and nervous tissue, initially the bead-reference region is approached, images are taken, automatically processed by rapidSTORM, and the mean z values for the beads are extracted. After that, the Auto Focus module moves the objective through six focus positions (customized step size); for each position the mean z value is determined. Finally, the z position with a z value closest to the reference z value is selected. Next, STORM images of the preselected ROIs (five in our experiment design) are acquired. As long as the infrared diode-based focus stabilization stays within its working range of 50 μm (which is only exceeded if the glass bottom dish is strongly inclined toward the objective), the distance between the sets of beads and the particular ROI is not relevant for focus reliability.

**Imaging experiments on U2OS cells**. U2OS cells constitutively expressing the nuclear pore component Nup133 tagged with Ypet were successively stained (Fig. 2a, Supplementary Table 1) with Alexa Fluor 647 conjugated nanobodies against GFP that bind Ypet (SR1/Ch1), phalloidin binding actin (SR2/Ch1), WGA-CF680 binding sialic acid and N-acetylglucosaminyl sugar residues at the plasma membrane, at the Golgi *trans* face and in the center of nuclear pores (SR2/Ch2), Alexa Fluor 647 conjugated clathrin antibodies (SR3/Ch1), anti-tubulin (SR4/Ch1), anti-paxillin (SR4/Ch2), anti-vimentin (SR5/Ch1), cis-Golgi recognizing anti-GM130 (SR5/Ch2), anti-fibrillarin (SR6/Ch2), mitochondria-specific anti-Tom20

(SR7/Ch1), anti-lamin A/C (SR8/Ch1), ER recognizing anti-PDI (SR8/Ch2) antibodies, as well as early endosome anti-Rab5 (SR9/Ch2) and anti-EEA1 (SR10/Ch1) antibodies, and a $JF_{646}$-Hoechst conjugate that reversibly binds DNA (SR11/Ch1). In addition to this representative experiment, another three experiments were carried out with similar designs and outcomes.

**Imaging experiments at the calyx of Held giant synapse**. Tissue was labeled with antibodies against bassoon (SR1/Ch1) and homer1/2/3 (SR1/Ch2), phalloidin Alexa Fluor 647 (SR2/Ch1), WGA-CF680 (SR2/Ch2), Alexa Fluor 647 labeled CHC17 primary antibody (SR3/Ch1), and antibodies against MAP2 (SR4/Ch1) VGlut1 SR4/Ch2), GM130 (SR5/Ch2), Rab3a (SR6/Ch1), Tom20 (SR6/Ch2), α-tubulin (SR7/Ch2), γ-actin (SR8/Ch1), synaptophsin 1 (SR8/Ch2), α/β-synuclein (SR9/Ch1), and PDI (SR9/Ch2), as well as $JF_{646}$-Hoechst conjugate that reversibly binds DNA (SR10/Ch1) (Fig. 3c, Supplementary Table 2). In addition to this representative experiment, another six experiments were carried out with similar designs and outcomes.

**SMLM image reconstruction and registration**. Directly after SMLM acquisition, employing a Python script, image frames were automatically redirected to rapid-STORM software for single-molecule localization and image reconstruction. For subsequent SMLM processing steps, including spectral demixing, drift correction, and 2D as well as 3D rendering, the Post-Processing software was developed. To allow custom data processing, we complemented the software by a GUI with modular design. Fast and standardized image analysis was possible using the batch processing function. SMLM images from consecutive staining rounds were registered based on fluorescent bead fiducials (at least five). SMLM images were linearly transformed based on mean distances along x and y axes between corresponding fiducials from different imaging rounds.

To determine the linear registration error, we first calculated the distances ($\Delta x$) between x coordinates of each bead i in an image from round A and bead i in an image from round B:

$$\Delta x_i = x_{A,i} - x_{B,i}.$$

Subsequently, based on $\Delta x$ of all beads n, we calculated the registration error (RE) along the x axis:

$$RE_x = \frac{1}{n} \sum_{i=1}^{n} \left| \overline{\Delta x} - \Delta x_i \right|$$

with $\overline{\Delta x}$ being the average $\Delta x$ over all beads. RE along the y axis was calculated analogously.

Fiducial coordinates were semi-automatically determined in ImageJ[50] and imported to Excel (Microsoft) where $RE_x$ and $RE_y$ were calculated as described above and visualized as a heatmap. For registration with image from round A, images from round B were linearly transformed by $\overline{\Delta x}$ and $\overline{\Delta y}$.

**Analysis of protein distribution at the calyx synapse**. The protein distribution within the calyx of Held was analyzed using discrete line profiles manually drawn in ImageJ (Supplementary Fig. 12). Initially, the tightly apposed pre- and post-synaptic membranes labeled by WGA were traced by the "segmented line selection tool". Using a custom-written ImageJ macro, the line was automatically interpolated, split into 1 μm segments, and rotated by a 90° angle. This ensured an orientation of line profiles perpendicular to the synaptic membrane. The line selections were elongated and the line thickness was set to 700 nm. To ensure reliable analysis, we considered contiguous membranes, which we defined as a clearly discernible WGA staining at both membranes (outer and inner) of the synapse that is traceable in at least 70% of the analyzed selection. Correct identification of synaptic membranes was additionally confirmed by AZ and SV markers. For all calyx segments that passed our criteria, multi-line plots were automatically generated and saved as tables. Data tables from five experiments (with three calyces per experiment and 4–16 line profiles per calyx; for n numbers see also Supplementary Table 5) were collected and imported to Matlab for further analysis. We define the outer and inner (synaptic) borders of the calyx as presynaptic membrane not facing the postsynaptic cell and presynaptic membrane facing the postsynaptic cell, respectively. The pre- and postsynaptic membranes constituting the inner border cannot be discriminated because they are separated by <20 nm in aldehyde-fixed tissue (unpublished electron microscopic measurements). The calyx borders were approximated by fitting the line profiles with the Gaussian function (Supplementary Fig. 12a). The mean distance between the outer and the inner calyx border was calculated from the whole WGA data set. All line profiles of the WGA signal and of all other proteins analyzed were scaled and aligned according to the borders of an average calyx. The difference between the calyceal distribution of α-tubulin, F-actin, and VGlut1 was quantified by localizing their intensity peaks using the Gaussian function. Statistical significance was evaluated by one-way analysis of variance followed by Bonferroni post hoc test.

For the analysis of AZ-positive vs. AZ-negative calyx segments, 300 nm thick-line profiles were laid across two AZ-containing calyx regions (characterized by dense bassoon signal) and two regions free of AZs per calyx (Supplementary Fig. 12b, c). In total, 13 calyces from five independent experiments were used for this analysis (for n numbers, see also Supplementary Table 5). The same data set

was used for the colocalization analysis. In this case, line profiles were manually restricted to the calyx and transformed to area selections. Correlation between all possible combinations of proteins within these selections was automatically (ImageJ macro) determined using the Pearson's coefficient. Data were exported to Matlab and extracted to colocalization matrices, which in turn have been further processed, averaged, and compiled as a heatmap in Excel. Colocalization data of selected pairs of proteins in AZ-positive vs. AZ-negative regions were statistically analyzed by unpaired two-tailed Student's t test.

**Reporting summary**. Further information on research design is available in the Nature Research Reporting Summary linked to this article.

## Data availability
The source data underlying Fig. 3g and Supplementary Figs. 5a–c, 6a, 8b–d, 9a, c, e, f, 10a, b, d are provided as a Source Data file. All data that support the findings of this study are available from the corresponding author upon reasonable request.

## Code availability
Microscope Control software with integrated Experiment Editor software (custom code: https://github.com/fherrmannsdoerfer/MicroscopyControl2), Chronos 4.8.0 (commercial). Plugin for Chronos (custom code: https://github.com/fherrmannsdoerfer/Chronos_Plugin).

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

## Acknowledgements

The purchase of the PAL3-pipetting robot was supported by the CellNetworks Excellence Cluster (EXC 81). We gratefully acknowledge the data storage service SDS@hd supported by the Ministry of Science, Research and the Arts Baden-Württemberg (MWK), and the German Research Foundation (DFG) through grant INST 35/1314–1 FUGG and INST 35/1503–1 FUGG. We thank Heinz Horstmann for training M.K. in cryo-sectioning of MNTB tissue, Claudia Kocksch for excellent technical assistance, and Carlo Beretta for help with image analysis. We also thank M. Simonetti (R. Kuner laboratory) for the Vimentin antibody, C. Spahn for the CHC17 antibody, and Luke Lavis for $JF_{646}$-Hoechst.

## Author contributions

M.K. designed and performed all experiments, developed computational image analysis tools, analyzed all data, and generated all images. F.H. with assistance of M.K. built the maS$^3$TORM setup and developed the acquisition interface "Experiment Editor". F.H. designed and developed the post-processing software for STORM images. S.S. performed and analyzed control experiments and produced Supplementary Movie 1. V.V. devised the concept of automating the STORM setup, conducted preliminary tests with a demo version of the robot, and pretested elution and bleaching protocols. M.H. and T.K. contributed to experimental design, data analysis, and supervised this work. M.K., M.H., and T.K. wrote the manuscript. All authors edited and approved the manuscript.

## Competing interests

The authors declare no competing interests.
