## [Peer Review File · Nature Communications]

Reviewers' comments:

Reviewer #1 (Remarks to the Author):

In this manuscript Thomas Kuner and collaborators introduce a very interesting technique for the repeated staining and STORM imaging of biological specimens. The technique is based mainly on immunostainings using antibodies, which are analyzed using STORM, and are then bleached and/or chemically eluted, to enable renewed immunostaining with another set of antibodies. In principle one could apply this tool for very many rounds of staining, thereby resulting in strong multiplexing.

The technology seems robust, and should enable many interesting analyses. The presentation of the results is convincing.

One control, however, seems to be missing. It would be important to quantify the reproducibility of this approach, by staining, imaging and eluting repeatedly the same epitope. This would verify whether the procedure affects the samples, and results in progressively poorer stainings, after each round of analysis, or whether it is harmless for the samples.

Reviewer #2 (Remarks to the Author):

Klevanski et al. proposes a nice piece of work exploring sequential labelling for SMLM. The capacity of imaging many structures of interest at the molecular scale in a single cell and correlate assemblies is a long-sought after goal in microscopy and molecular and cellular biology. The work shows a pipeline for sequential labelling protocols and SMLM imaging in a single sample, allowing the use of the same antibody species multiple times. This approach has the advantage of allowing potentially an unlimited number of targets. They demonstrate the work on cell culture (showing common labels such as cytoskeleton and NPC) and in neuronal tissue (looking at pre and post synaptic common molecular assemblies).

This approach was previously demonstrated by the Yi et al. 2016 (madSTORM), but not in an automated fashion. The capability for automation was shown by Almada et al. 2018 (NanoJ-Fluidics), but little work was done by these authors on optimising protocols of sequential labelling. Although it is an achievement to interface programmatically two pieces of hardware so different (one commercial and one based on open source microscopy), the automation approach taken in this manuscript is specifically optimised for a combination of software and hardware that is not available to the general community and therefore is less applicable. Therefore, the main interest in this work is in establishing optimised protocols of sequential labelling and imaging for common and important targets in cells and tissue. Although all the approaches described here are scientifically valid and well-studied, I think the author have not pushed the optimisation and the methods to a level granting publication in Nature Communications, considering that none of the ideas described here are new in themselves. However, if the authors judge possible to address several major concerns about the optimisations and generalization of the method, I would be happy to reconsider my opinion.

Major points:

The major concern with these sequential labelling approaches is that there may be cumulative damage to the sample across the rounds of labelling, as a consequence of high intensity illumination and harsh protocol of Ab elutions. Two main issues may occur: the stripping off of molecules of interest over time and therefore losing structure (which may also vary between different molecules of interest/Ab combination) and loss of general integrity of the sample (e.g. collapsing of structures, breaking of molecular bonds). From this point of view (that of technology development), the most important piece of data is on SuppFig 3 and SuppNote1 where a detailed analysis of the effect of bleaching and elution is performed on Tom20 labels. The experiments are

well described and the conclusions appear robust and are interesting. However, I see several limitations:

- There needs to be a more detailed description of the quantification performed here (counting several localizations in a ROI is a very basic level of quantification and is not normalised to surface area, choice of ROIs etc.)
- The number of locs does not reflect potential loss of structural integrity. Representative images should be shown and compared across the rounds. This spatial comparison should be quantitatively estimated by image metrics.
- The analysis is only performed for one single antibody. Might other type of antibody react differently to bleaching and elution? I would certainly expect so.
- Importantly, it only shows the results after a single round of bleaching/elution. I suspect that 10 rounds will have a cumulative effect that may substantially affect the sample. Could the effect of consecutive elution/bleaching, re-labelling of the same structure be tested over many (10?) rounds to demonstrate the absence of loss of signal/structure on image-based metrics?

Sup Fig 4 explores some of these ideas with some quantitative metrics. However, by using the data taken from Figure 2 and not bespoke experiments testing these ideas, the conclusions cannot be drawn very conclusively in my understanding. How was the CT metric calculated here? Since the ground truth of the second label is unknown. Also, different labels were used in the first case (from AF647 to AF680), which after demixing might artificially improve the measured CT. Also 6.5% CT seems to be in contradiction with the 1.8% observed in SupFig3.

The CT observed in further analysis here show that there is indeed a strong dependence on Ab types and that the consecutive rounds of labelling can contain a significant amount of labels from previous rounds. These are in direct contradiction with the broad statement made by the authors in the main manuscript: "This shows that our strategies for signal removal cause minimal crosstalk between staining rounds (for examples with other targets, see Supplementary Fig. 4a-d), a prerequisite for high-fidelity multiplexed imaging."

Also, the statement "staining can be maintained while many additional staining rounds are added" is really unclear to me. It seems to be referring to the fact that molecules of interests can still be labelled efficiently after multiple rounds of bleaching/labelling. But the figures shown in SuppFig4 show a ~63% loss of labels at round 8, this seems really poor to me. How to expect that images obtained towards the end of the acquisition to have high fidelity towards the structure of interest? In a macromolecular assembly this would represent a very high level of false negative.

It is also not clear to me how it's possible to compare labelling efficiency after many rounds to a different cell at round 1. It should be done on the same cell (as I suggested earlier) or done in a normalised manner on high statistics.

On Figure 2, the authors show beautiful NPC and WGA 2-color reconstructions from early rounds and poor quality cytoskeleton images at later round (panel e makes me question the validity of a ~7nm localization precision here). This raises the question of: Does the imaging get worse as the rounds of labelling advance? How can the authors explain this?

Could the authors reverse the labelling rounds and still obtain these images of NPCs at round 10? This really is an important aspect of the approach itself at a fundamental level. Additionally, no structural metrics such as sizes and distances between molecular assemblies are provided to validate the SRM imaging.

"Two channels (Ch) can be detected per SR, but not always each channel can be used for final analysis. In addition, to get optimal signal for some sensitive labels (e.g. GFP nanobodies against the Ypet-tag of Nup133), it was beneficial to not stain targets in the second channel." The authors should describe what they mean and why there are some optimal configurations and why not using 2 channels every time (which could limit the necessary rounds of elution/bleaching). The impact and importance of this work lie in establishing the rationale behind optimised protocols of sequential labelling and therefore the readers will be interested in a thorough and in depth

description of how to go about designing their own protocols. The registration precisions are quite poor here (up to 68 nm errors). This is a real technical limitation for the method and should be tackled by the author. They mention tethering the beads to prevent them from moving, this is a good idea.

Figure 3 shows some work in neuronal tissue taking the same approach. Does the labelling and effect of elution work as well as in cells in tissue? Some test should be done to show that. It's unlikely that the protocols do not need to be adjusted for optimal elution. I also have to admit that the images shown here are not very impressive for SMLM, although I admit it is not as easy to get SMLM working well in tissue.

How were the errors of registration precision calculated between each round to compute the heatmaps? More details are necessary here. Supp Note 2 is an important discussion about long term registration in this approach but does not describe the method. This should be discussed in the main text in my opinion.

The quantitative analysis of protein distribution across the gap seems well done and quantitative. But this paper is primarily about the development of the technology.

Conclusion: "multiplexing by optimized re-staining protocols". It is arguable that the protocols are optimised as no other methods is shown, compared and tested. Not even varying amount of time for elution or elution buffer composition. The buffer is currently very close to that used in Collman et al. 2015.

Minor points:

Intro: a deeper description of the sequential labelling methods demonstrated to date would be beneficial to the reader. A number of seminal papers are mentioned but not described in sufficient details. The madSTORM work clearly sets up some ground work on this type of sequential labelling. The difference and advantages beyond the automation should be presented and discussed. Similarly a short overview of approaches of unmixing of far-red fluorophores would also be great, as it is central to such methods.

The idea of using a set of beads to refocus before moving onto the ROIs of interest is interesting. But just how far away from this FOV of beads are all the ROIs? Is there a maximum distance before the focus position obtained this way becomes unreliable?

The issue comes with the use of the beads as fiducials in the acquisition and registration across rounds of imaging. The beads seem to only be useable for 11 rounds. Why not use nanogold or nanodiamond as demonstrated by Yi et al. (madSTORM)?

"Therefore, to re-establish a focal plane in late imaging rounds comparable to that from the initial round, the focus would have to be placed more and more deeply into the sample, resulting in less focused fiducial beads, and thus, less precise fitting and a higher registration error." This does not make sense to me in an automated approach as described here as the focus will be fixed with respect to the position of the beads used for focussing, so therefore the structure of interest will move out of the focus and not the fiducial beads. Could the authors please clarify?

SI SupFig 1: "While the Microscope Control software is directly addressed by the Experiment Editor software, the communication between Experiment Editor and the pipetting robot is realized indirectly (depicted by the dashed line) via an exchange folder that is constantly checked and modified by the custom-written plugin for Chronos as well as by the Experiment Editor." More details are necessary about the protocol of communication between the robot and the experiment editor. What is the exchange folder protocol? How "real-time" is this? An idea of the efficiency of such protocol should be discussed. Also, why choose this approach and not any other protocol? Readers wanting to replicate the work or adapt it to their hardware would be interested in

understanding the rationale behind these decisions.

In the supp tables the concentrations of JF dyes are given in nm and not nM (typo).

SuppFigure 6 seems to have some color-coded depth missing or information missing on what the colors mean.

"To further improve our computational efficiency, fiducial-based registration could be implemented into the post-processing software." I am confused by this statement, I was under the impression that fiducial markers were used to register the different structures?

"a precision of approximately 10-25 nm" in cells, registration errors go up to ~68 nm, so this is somewhat an overstatement.

Response to reviewers

Reviewers' comments:

Reviewer #1 (Remarks to the Author):

In this manuscript Thomas Kuner and collaborators introduce a very interesting technique for the repeated staining and STORM imaging of biological specimens. The technique is based mainly on immunostainings using antibodies, which are analyzed using STORM, and are then bleached and/or chemically eluted, to enable renewed immunostaining with another set of antibodies. In principle one could apply this tool for very many rounds of staining, thereby resulting in strong multiplexing.

The technology seems robust, and should enable many interesting analyses. The presentation of the results is convincing.

Response: We thank the reviewer for the positive assessment of our work.

(1) One control, however, seems to be missing. It would be important to quantify the reproducibility of this approach, by staining, imaging and eluting repeatedly the same epitope. This would verify whether the procedure affects the samples, and results in progressively poorer stainings, after each round of analysis, or whether it is harmless for the samples.

Response: We thank the reviewer for this suggestion, which we agree is an important control. We performed new experiments and repetitively stained, bleached, eluted and imaged mitochondria (Tom20) in cells and tissue over 10 rounds. The data is presented in the new Supplementary Figure 8. We fitted a regression line to the data and found an average decrease of the signal of 2.7% (cells) and 4.1% (tissue) per imaging round (Supplementary Fig. 8a,c). The integrity of the structure was determined by cross-correlation analysis, revealing no progressive structural damage neither in cells nor in tissue (Supplementary Fig. 8e, f). These results are now mentioned in the main text (p.3, 4).

In conclusion, these data demonstrate that our procedure can be applied for many rounds without suffering from a major loss of staining or sample destruction. Moreover, the experiment presented here rather reflects a worst-case-scenario, because a typical experiment would not include an elution step in every round (see Supplementary Tables 1,2; Supplementary Note 2).

Reviewer #2 (Remarks to the Author):

Klevanski et al. proposes a nice piece of work exploring sequential labelling for SMLM. The capacity of imaging many structures of interest at the molecular scale in a single cell and correlate assemblies is a long-sought after goal in microscopy and molecular and cellular biology. The work shows a pipeline for sequential labelling protocols and SMLM imaging in a single sample, allowing the use of the same antibody species multiple times. This approach has the advantage of allowing potentially an unlimited number of targets. They demonstrate the work on cell culture (showing common labels such as cytoskeleton and NPC) and in

neuronal tissue (looking at pre and post synaptic common molecular assemblies).

Response: We thank the reviewer for the positive and very detailed assessment of our work that helped to significantly improve the quality of our manuscript.

(2) This approach was previously demonstrated by the Yi et al. 2016 (madSTORM), but not in an automated fashion. The capability for automation was shown by Almada et al. 2018 (NanoJ-Fluidics), but little work was done by these authors on optimising protocols of sequential labelling.

Response: We thank the reviewer, and we agree that the manuscripts mentioned did not optimize multiplexed imaging. We optimized staining, elution, and fluid exchange using robotics, which compared to nanofluidics has a negligible dead volume and a theoretically unlimited number of channels (in our settings, we have 162 reservoirs available; commercial nanofluidics currently operate up to 15 channels (Bruker)). Yet most importantly, we demonstrate the usefulness of the method by showing 3D, 16-color imaging in neuronal tissue, and revealing novel findings about the nano-architecture of proteins in synapses.

(3) Although it is an achievement to interface programmatically two pieces of hardware so different (one commercial and one based on open source microscopy), the automation approach taken in this manuscript is specifically optimised for a combination of software and hardware that is not available to the general community and therefore is less applicable.

Response: We agree with the reviewer that applicability is an important point. Yet, we are convinced that our solution is generally applicable, and could be established even by less specialized labs. All the software is available (acquisition, robot control, analysis). As mentioned by the reviewer, the pipetting robot is commercially available. All microscopic components can be purchased. Due to the modular principle of μ Manager, hardware components can be substituted by equivalent devices without changing the code as long as they can be operated by the same μ Manager driver. Individual components can also be excluded, which requires minimal changes of the code. We now included extended information on the communication between the robot and the microscope (Supplementary Figure 3). We are more than happy to support future applicants of our system.

(4) Therefore, the main interest in this work is in establishing optimised protocols of sequential labelling and imaging for common and important targets in cells and tissue. Although all the approaches described here are scientifically valid and well-studied, I think the author have not pushed the optimisation and the methods to a level granting publication in Nature Communications, considering that none of the ideas described here are new in themselves. However, if the authors judge possible to address several major concerns about the optimisations and generalization of the method, I would be happy to reconsider my opinion.

Response: We thank the reviewer for this frank and critical assessment. However, we are not aware of any work demonstrating 3D, 16-color, super-resolution imaging of protein nano-patterns in tissue.

Automation of high-throughput multiplexing for super-resolution setups is a non-trivial task that until now remained unsolved. Main obstacles were (1) spatial restrictions that precluded the simultaneous usage of large super-resolution setups and bulky liquid exchange devices, (2) the limited number of channels in micro-fluidic chambers, and (3) drift-sensitivity of the sample. Our main achievement was not only to employ a pipetting robot for liquid exchange at a microscope (which is a novel idea) and to interconnect its hard- and software with the STORM setup, but also to curb sample drift and provide a solution for auto-refocusing.

At the same time, we want to emphasize that this paper is not solely describing a method for multiplexing. We have presented (in Figures 3 and 4) novel insights into the nanoscopic organization of proteins in and next to glutamatergic synapses. These data substantially enrich fundamental neurobiological knowledge. We believe that automated super-resolution multiplex solutions open new avenues to consistent and systematic findings on large scale in different biological fields.

We have performed additional experiments and added these results on optimizing the elution buffer, elution time, and bleaching time. These data are shown in Supplementary Figures 4 and 8. Other data was reworked for clarity and new analyses were integrated (e.g. Supplementary Figures 5, 6, 7, 9, 10).

Major points:

(5) The major concern with these sequential labelling approaches is that there may be cumulative damage to the sample across the rounds of labelling, as a consequence of high intensity illumination and harsh protocol of Ab elutions. Two main issues may occur: the stripping off of molecules of interest over time and therefore losing structure (which may also vary between different molecules of interest/Ab combination) and loss of general integrity of the sample (e.g. collapsing of structures, breaking of molecular bonds).

From this point of view (that of technology development), the most important piece of data is on SuppFig 3 and SuppNote1 where a detailed analysis of the effect of bleaching and elution is performed on Tom20 labels. The experiments are well described and the conclusions appear robust and are interesting.

However, I see several limitations:

5.1 There needs to be a more detailed description of the quantification performed here (counting several localizations in a ROI is a very basic level of quantification and is not normalised to surface area, choice of ROIs etc.)

5.2 The number of locs does not reflect potential loss of structural integrity. Representative images should be shown and compared across the rounds. This spatial comparison should be quantitatively estimated by image metrics.

Response:

5.1 We have added the requested information to Supplementary Figure 5 by integrating panel b, showing an overview image with the regions selected for analysis highlighted, and the caption of Supplementary Figure 5. We wish to emphasize that the analysis was ROI-based and image-based. Images were rendered in a way that pixel intensity allows to directly extract information about the number of localizations in a certain area. For the analysis we considered only regions fully covered by Tom20 signal (now highlighted as magenta boxes in Supplementary Fig. 5b), as well as regions of background (yellow boxes), which addresses the concerns of normalization.

5.2 This criticism was also raised by reviewer #1. We now show images in Supplementary Figure 6. We performed new experiments and repetitively stained, bleached, eluted and imaged mitochondria (Tom20) in cells and tissue over 10 rounds. The data is presented in the new Supplementary Figure 8. We fitted a regression line to the data and found an average decrease of the signal of 2.7% (cells) and 4.1% (tissue) per imaging round (Supplementary Fig. 8a,c). The integrity of the structure was determined by cross-correlation analysis, revealing no progressive structural damage neither in cells nor in tissue (Supplementary Fig. 8e, f). These results are now mentioned in the main text (p.3, 4).

(6) The analysis is only performed for one single antibody. Might other type of antibody react differently to bleaching and elution? I would certainly expect so.

Response: We agree with the reviewer, but at the same time, we wish to emphasize that a similar analysis of all antibodies would be beyond the scope of this manuscript and not feasible within a reasonable time frame: a comparably detailed analysis requires 18 sub-experiments and would take about two months for every additional protein to be tested. We chose a well-characterized structure for our analysis, mitochondria (other less defined structures also might be more difficult to analyze). We show, on the basis of small data sets, the analysis of other antibodies in Supplementary Figure 7, panels b – d. We also comment on this point to make the prospective user aware of the fact that in principle, each antibody needs to be carefully characterized (p. 4, bottom).

(7)- Importantly, it only shows the results after a single round of bleaching/elution. I suspect that 10 rounds will have a cumulative effect that may substantially affect the sample. Could the effect of consecutive elution/bleaching, re-labelling of the same structure be tested over many (10?) rounds to demonstrate the absence of loss of signal/structure on image-based metrics?

Response: We performed 10 rounds of imaging in the course of the revision, analyzed fluorescence signal, and determined cross-correlation coefficients, both for cells and in tissue (Supplementary Fig. 8). This result very convincingly shows that the loss in signal intensity is 2.7-4.1% per round in cells and tissue.

(8) Sup Fig 4 explores some of these ideas with some quantitative metrics. However, by using the data taken from Figure 2 and not bespoke experiments testing these ideas, the conclusions cannot be drawn very conclusively in my understanding. How was the CT metric calculated here? Since the ground truth of the second label is unknown. Also, different labels were used in the first case (from AF647 to AF680), which after demixing might artificially improve the measured CT. Also 6.5% CT seems to be in contradiction with the 1.8% observed in SupFig3.

Response: We added information in the figure caption (now Supplementary Fig. 7, SI: pp 7-8) how cross-talk was determined. Indeed, we compared labels with different fluorophores across rounds (first panel row of Suppl. Fig. 7a). This was the only possibility to estimate cross-talk from consecutive rounds as in our experimental design we intentionally alternated fluorophores of secondary antibodies against the same species (see new Supplementary Note 2). As for

demixing, a 690 nm dichroic mirror is used, by switching from a fluorophore emitting at 647 nm to 680 nm we rather overestimate CT. We added sentences in the figure caption (SI: p. 11) that state that in certain cases cross-talk and Pearson's r coefficient might be influenced by the fluorophore and depend on the target structures.

The reviewer may have accidentally compared the cross-talk of tubulin data (now Supplementary Fig. 7a, panel row 1: 6.4%) with Tom20 data (now Supplementary Fig. 5a, test condition 6: 1.8%), however, this does not represent a contradiction.

(9) The CT observed in further analysis here show that there is indeed a strong dependence on Ab types and that the consecutive rounds of labelling can contain a significant amount of labels from previous rounds. These are in direct contradiction with the broad statement made by the authors in the main manuscript: "This shows that our strategies for signal removal cause minimal crosstalk between staining rounds (for examples with other targets, see Supplementary Fig. 4a-d), a prerequisite for high-fidelity multiplexed imaging."

Response: We agree with the reviewer and now modified the sentence mentioned (p. 7). Multi-labeling experiments need to be designed considering several experimental parameters. We now provide a set of general rules to guide the user how to optimally design such an experiment (Supplementary Note 2).

(10) Also, the statement "staining can be maintained while many additional staining rounds are added" is really unclear to me. It seems to be referring to the fact that molecules of interests can still be labelled efficiently after multiple rounds of bleaching/labelling. But the figures shown in SuppFig4 show a ~63% loss of labels at round 8, this seems really poor to me. How to expect that images obtained towards the end of the acquisition to have high fidelity towards the structure of interest? In a macromolecular assembly this would represent a very high level of false negative. It is also not clear to me how it's possible to compare labelling efficiency after many rounds to a different cell at round 1. It should be done on the same cell (as I suggested earlier) or done in a normalised manner on high statistics.

Response: We agree with the reviewer and we apologize for generating confusion. We indeed compared intensity data from different experiments (n = 1). We now performed a systematic analysis of how staining is maintained over several rounds, added new experimental data and analyses to the revised manuscript (Supplementary Fig. 8) and altered the manuscript accordingly (p. 4, bottom).

(11) On Figure 2, the authors show beautiful NPC and WGA 2-color reconstructions from early rounds and poor quality cytoskeleton images at later round (panel e makes me question the validity of a ~7nm localization precision here). This raises the question of: Does the imaging get worse as the rounds of labelling advance? How can the authors explain this?

Response: We addressed this criticism by quantifying spatial resolution over 10 imaging rounds using decorrelation analysis (new Supplementary Fig. 9b, d; main text pp. 5, 6), strongly supporting the previous (now Supplementary Fig. 9a) data on localization precision. In addition, we analyzed the image quality (intensity and cross-correlation of structures) and show these results in the new

Supplementary Figure 8 (in particular panels e and f; main text p. 5, 1st paragraph). The results demonstrate that resolution and cross-correlation remain rather stable over 10 imaging rounds. Hence, imaging does not get worse as the rounds of labeling advance.

The signal loss of 2.7-4.1% per imaging round shown in Supplementary Fig. 8a-d is presumed to arise from epitope modification or loss. For most targets with a high number of epitopes and labels with high labeling efficiency, this moderate signal loss can be neglected. However, especially targets with a low number of epitopes (e.g. Nup133 with 4 epitopes per NPC subunit) should be imaged in early multiplexing rounds. This also becomes apparent from the new experiment (see Supplementary Fig. 8g) where Nup133 was labeled after 10 rounds of elution and bleaching resulting in sparse labeling. In general, structural preservation of many different types of proteins is a challenging task. We now formulated a list of rules to guide the experimenter (see new Supplementary Note 2). Depending on the scientific question and the targets of interest, experimenters need to find the best compromise of preserving different targets and design their priority order of rules accordingly.

(12) Could the authors reverse the labelling rounds and still obtain these images of NPCs at round 10? This really is an important aspect of the approach itself at a fundamental level.

Response: We designed the sequence of labeled targets based on a priori knowledge on epitope density and abundance, as well as other experimental parameters. As discussed in Supplementary Note 2, the order of labeling steps is essential for achieving an optimal multiplex labeling outcome. Hence, experimental sequences designed according to these rules cannot be reversed. When the experiment is poorly designed, the image quality obtained in the final rounds may in fact suffer. For example, when considering NPCs, the loss of epitopes after 10 rounds of imaging (see new Supplementary Figure 8) paired with the limited accessibility of the epitopes in this densely packed structure will affect image quality (e.g. Nyquist criterion). As suggested by the reviewer, we included a new experiment showing NPCs visualized after 10 labeling rounds (Supplementary Fig. 8g). However, in line with the considerations discussed above, eightfold symmetry was not detectable. Therefore, NPCs and target proteins/structures having similar features should be imaged at the beginning of an experiment. We developed a set of rules for the design of multiplex labeling experiments that makes the user aware of these issues and supports the best possible outcome (new Supplementary Note 2). Nevertheless, as discussed in the previous comment #11, this is not a fundamental problem for the approach as such, because structural integrity is maintained.

(13) Additionally, no structural metrics such as sizes and distances between molecular assemblies are provided to validate the SRM imaging.

Response: We now addressed this point and added decorrelation analysis of a whole multiplexing experiment (supporting localization precision data) in the new Supplementary Figure 9. We now also quantified diameters of NPCs based on intensity profiles resulting in the expected values (Supplementary Fig. 9c).

(14) "Two channels (Ch) can be detected per SR, but not always each channel can be used for final analysis. In addition, to get optimal signal for some

sensitive labels (e.g. GFP nanobodies against the Ypet-tag of Nup133), it was beneficial to not stain targets in the second channel." The authors should describe what they mean and why there are some optimal configurations and why not using 2 channels every time (which could limit the necessary rounds of elution/bleaching). The impact and importance of this work lie in establishing the rationale behind optimised protocols of sequential labelling and therefore the readers will be interested in a thorough and in depth description of how to go about designing their own protocols.

Response: The reason of using only one channel in the Nup133 imaging round was the low number of epitopes and the limited accessibility of epitopes. To avoid any background contribution from demixing (although very low), we decided to perform one-channel imaging. Other reasons for one-channel imaging are antibodies producing very low signals and the limited availability of orthogonal antibodies for specific targets. We included these considerations to the set of rules for the design of optimal multiplex experiments (new Supplementary Note 2) and referred to this note (p. 23, Legend to Fig. 2).

(15) The registration precisions are quite poor here (up to 68 nm errors). This is a real technical limitation for the method and should be tackled by the author. They mention tethering the beads to prevent them from moving, this is a good idea.

Response: The reviewer is concerned about a registration error that occurred in imaging round #11, and in cells (Figure 2b), which is the highest value and well above the average values of 31 nm for cells and 17 nm for tissue. The exceptionally low registration precision of 68 nm can simply be explained with the fact, that in the last imaging round #11, we used the DNA-staining Janelia fluorophore Hoechst-JF646 in PAINT mode, which requires to keep the dyes in solution. The increase in background reduced the number of beads that could be used for registration, and worsened their localization precision at the same time. Triggered by the reviewer's question, we now also found that the registration error for the last imaging round #10 of the multiplex experiment in tissue (Figure 3d) was inadvertently missing and now added the missing values. Furthermore, JF646-Hoechst staining resulted in substantially higher values for the registration error in this experiment, supporting our explanation above. We now included these considerations in the new Supplementary Note 2; SI: p. 23). In general, we obtained better results in tissue, which is in part explained by the better structural robustness and adhesion of tissue compared to cells (please see new Supplementary Note 3; SI: pp. 23, 24).

(16) Figure 3 shows some work in neuronal tissue taking the same approach. Does the labelling and effect of elution work as well as in cells in tissue? Some test should be done to show that. It's unlikely that the protocols do not need to be adjusted for optimal elution. I also have to admit that the images shown here are not very impressive for SMLM, although I admit it is not as easy to get SMLM working well in tissue.

Response: We performed additional experiments to compare the quality of re-staining and imaging, and obtained very similar results for cells and tissue using the same conditions (Supplementary Figure 8). While thick tissue sections used

for conventional microscopy require special treatment, such as permeabilization and extensive blocking of high background, for super-resolution imaging we use thin tissue sections (400 nm) that do not require permeabilization (because due to the thin section thickness, all cellular compartments are cut open and therefore accessible to the antibodies) and can otherwise be treated like cells (see our previous work: (Kempf *et al.*, 2013; Körber *et al.*, 2015)). We emphasized that in the revised manuscript (p. 11).

Regarding the quality of tissue images, we are not aware of any comparable data in the literature. To support this statement, for the revision process, we assembled a list of publications using super-resolution SMLM in tissue (see Table 1 for revision).

(17) How were the errors of registration precision calculated between each round to compute the heatmaps? More details are necessary here. Supp Note 2 is an important discussion about long term registration in this approach but does not describe the method. This should be discussed in the main text in my opinion.

Response: We agree with the reviewer that these details are necessary. We now added missing information about the calculation of the registration precision to the methods section.

However, to preserve readability of the main text and given the complexity and level of detail that needs to be covered, we would prefer to keep this information in the methods section (main text: p. 15) and Supplementary Note 3 (SI: pp. 23, 24). Nevertheless, we emphasized the importance of this point in the main text and explicitly refer to those sections.

(18) The quantitative analysis of protein distribution across the gap seems well done and quantitative. But this paper is primarily about the development of the technology.

Response: The paper is about technology development, but we consider it crucial to illustrate the applicability of such technology to relevant biological questions. Furthermore, our novel analysis also serves another purpose: it illustrates the overwhelming need for even more powerful analysis approaches, for example using convolutional neural networks, to detect hidden organizational principles in highly multiplexed protein distribution patterns.

(19) Conclusion: "multiplexing by optimized re-staining protocols". It is arguable that the protocols are optimized as no other method is shown, compared and tested. Not even varying amount of time for elution or elution buffer composition. The buffer is currently very close to that used in Collman *et al.* 2015.

Response: We rephrased the sentence mentioned by the reviewer and removed the term "optimized" to avoid misleading interpretations (main text: p. 8). What we meant with "optimized" refers to what we have now summarized in Supplemental Note 2 (designing an optimal order of staining rounds). To address the criticism raised by the reviewer, we now include additional experiments and analyses probing elution time, composition of the elution buffer, and bleaching time (new Supplementary Fig. 4). However, the parameter space

is vast and therefore can only be probed within certain limits. These experiments show that our choice of parameters is well justified.

Minor points:

(20) Intro: a deeper description of the sequential labelling methods demonstrated to date would be beneficial to the reader. A number of seminal papers are mentioned but not described in sufficient details. The madSTORM work clearly sets up some ground work on this type of sequential labelling. The difference and advantages beyond the automation should be presented and discussed. Similarly a short overview of approaches of unmixing of far-red fluorophores would also be great, as it is central to such methods.

Response: We rephrased the introduction (p. 2) and added a more detailed description. We also include additional references on demixing for different fluorophores. For the revision process, we provide the reviewer with a table (Table 2 for revision) summarizing other methods and their achievements.

(21) The idea of using a set of beads to refocus before moving onto the ROIs of interest is interesting. But just how far away from this FOV of beads are all the ROIs? Is there a maximum distance before the focus position obtained this way becomes unreliable?

Response: We believe that our description of refocusing was insufficient and invited misunderstanding. The distance between the sets of beads and a particular ROI is not relevant – as long as the focus lock is kept. We rephrased this part of the methods section (main text: p. 13).

(22) The issue comes with the use of the beads as fiducials in the acquisition and registration across rounds of imaging. The beads seem to only be useable for 11 rounds. Why not use nanogold or nanodiamond as demonstrated by Yi et al. (madSTORM)?

Response: The bead performance in imaging round #11 (cells) was temporarily worsened by including an imaging round using the PAINT mode (see discussion above on the registration error for JF646, response to reviewer comment #15). The beads could likely be used for many more imaging rounds, if needed. Of course, other fiducial markers may also be used.

(23) “Therefore, to re-establish a focal plane in late imaging rounds comparable to that from the initial round, the focus would have to be placed more and more deeply into the sample, resulting in less focused fiducial beads, and thus, less precise fitting and a higher registration error.” This does not make sense to me in an automated approach as described here as the focus will be fixed with respect to the position of the beads used for focussing, so therefore the structure of interest will move out of the focus and not the fiducial beads. Could the authors please clarify?

Response: The reviewer is entirely right. Our statement was confusing as we missed to mention that in some rare cases (only for cell imaging) we had to manually readjust the focus to compensate for unexpected effects such as cell

detachment. We now clarified that in Supplementary Note 3 (SI: pp. 23, 24). We did not observe this in tissue, which we attribute to the better adhesion of slices to the glass support (which is also reflected in the better registration precision).

(24) SI SupFig 1: “While the Microscope Control software is directly addressed by the Experiment Editor software, the communication between Experiment Editor and the pipetting robot is realized indirectly (depicted by the dashed line) via an exchange folder that is constantly checked and modified by the custom-written plugin for Chronos as well as by the Experiment Editor.” More details are necessary about the protocol of communication between the robot and the experiment editor. What is the exchange folder protocol? How “real-time” is this? An idea of the efficiency of such protocol should be discussed. Also, why choose this approach and not any other protocol? Readers wanting to replicate the work or adapt it to their hardware would be interested in understanding the rationale behind these decisions.

Response: We generated a new Supplementary Figure 3 (SI: pp. 5, 6) which explains the communication between the robot and the experiment editor in detail.

(25) In the supp tables the concentrations of JF dyes are given in nm and not nM (typo).

Response: We thank the reviewer for spotting this mistake, and corrected it.

(26) SuppFigure 6 seems to have some color-coded depth missing or information missing on what the colors mean.

Response: We added a scale bar with color code for imaging depth in this figure (now Supplementary Fig. 10).

(27) “To further improve our computational efficiency, fiducial-based registration could be implemented into the post-processing software.” I am confused by this statement, I was under the impression that fiducial markers were used to register the different structures?

Response: We apologize for this confusing statement. We aimed at proposing a solution for automated registration, integrated in the post-processing software. Because this is not a point sufficiently important to make, we removed the sentence.

(28) “a precision of approximately 10-25 nm” in cells, registration errors go up to ~68 nm, so this is somewhat an overstatement.

Response: We thank the reviewer for pointing this out. Sentence rephrased.

References

Kempf, C. *et al.* (2013) ‘Tissue Multicolor STED Nanoscopy of Presynaptic Proteins in the Calyx of Held’, *PLoS ONE*. Edited by J.-P. Mothet, 8(4), p. e62893.

doi: 10.1371/journal.pone.0062893.

Körber, C. *et al.* (2015) 'Modulation of Presynaptic Release Probability by the Vertebrate-Specific Protein Mover', *Neuron*, 87(3), pp. 521–533. doi: 10.1016/j.neuron.2015.07.001.

Table 1 for revision: Super-resolution microscopy in neuronal tissue samples.

Super-resolution method	Nervous tissue	Quality	Multi-color	Targets	Publication
STORM					
3D dSTORM, spectral demixing & multiplexing	mouse CNS: MNTB synapse	LP xy ~ 7 nm, LP z ~ 30 nm	16 colors	see current paper	Klevanski et al.* (current not published paper)
3D STORM, activator-reporter pairs	mouse CNS: olfactory bulbs, cortex	LP xy ~14 nm, total convolved localization uncertainty: ~17 nm	3 colors	Homer1, Bassoon, Piccolo, Shank1, GluR1, Gaba _B R1, NR2B, CaMKII, PSD95	Dani et al. 2010
2D STORM	mouse CNS: hippocampus CA1		2 colors	Bassoon, GluR1/2	Beaudoin et al. 2012
dSTORM, spectral demixing	rat CNS: calyx of Held	SR xy ~28 nm	2 colors	membrane-bound GFP, Cytochrome C	Nanguneri et al. 2012*
dSTORM	Drosophila melanogaster PNS: neuromuscular junction	LP xy ~6-7 nm	1 color	Bruchpilot	Ehmann et al. 2014
dSTORM, spectral demixing	rat CNS: calyx of Held	LP xy ~7 nm, LP z ~30 nm	2 colors	Bassoon, VGlut1, Mover	Körber et al. 2015*
3D dSTORM	mouse CNS: hippocampus	LP xy ~6 nm, LP z ~41 nm	2 colors	CB1 receptor, Biocytin, Bassoon	Dudok et al. 2015
2D & 3D dSTORM	Drosophila melanogaster CNS: olfactory circuit		1 color	Bruchpilot, acetylcholine receptor Dα7, Synaptotagmin, Drep-2	Spühler et al. 2016
3D dSTORM	rat CNS: hippocampus	<10 nm		RIM1/2, PSD95	Tang et al. 2016
3D dSTORM	mouse CNS: hippocampus CA1		3 colors	GLT-1, Bassoon, Homer1	Heller et al. 2019
STED					
STED	mouse CNS: somato-sensory cortex	SR xy <70 nm	1 color	cytosolic dye	Berning et al. 2012
STED	rat CNS: calyx of Held	SR xy ~40 nm	2 colors	Synaptophysin, Rab3a, Synapsin, VGlut1	Kempf et al. 2013*
STED	mouse PNS: neuromuscular junction	SR xy ~40 nm	2 colors	Bassoon, Piccolo, P/Q-type VGCC	Nishimune et al. 2016
STED	mouse CNS: hippocampus	SR xy ~50 nm	1 color	GFP in pyramidal cells	Chéreau et al. 2017
3D STED	mouse CNS: hippocampus	SR xy ~60 nm, SR z ~160 nm	2 colors	extracellular space dye, cytosolic dye	Tønnesen et al. 2018
PALM					

COOL (PALM-related)	Drosophila melanogaster CNS: whole brain	SR xy ~20 nm, SR z ~1 μ m	1 color	cytosolic dye	Lin et al. 2019
PAINT					
Exchange-PAINT	mouse CNS: retina	subpixel precision xy ~5 nm, z ~75 nm	8 colors	SV2, GFAP, Cone arrestin, Chx10, Vimentin, Synapsin; non-paint: Lectin, DAPI	Wang et al. 2017

CNS = central nervous system, PNS = peripheral nervous system, MNTB = medial nucleus of the trapezoid body, SR = spatial resolution, LP = localization precision, RP = registration precision, (d)STORM = (direct) stochastic optical reconstruction microscopy, STED = stimulated emission depletion, PALM = photoactivated localization microscopy, COOL = confocal localization deep imaging with optical clearing, PAINT = point accumulation for imaging in nanoscale topography: *our laboratory

References:

- Dani, A., Huang, B., Bergan, J., Dulac, C. & Zhuang, X. Superresolution Imaging of Chemical Synapses in the Brain. *Neuron* 68, 843–856 (2010).
- Beaudoin, G. M. J. et al. Afadin, A Ras/Rap Effector That Controls Cadherin Function, Promotes Spine and Excitatory Synapse Density in the Hippocampus. *J. Neurosci.* 32, 99–110 (2012).
- Nanguneri, S., Flottmann, B., Horstmann, H., Heilemann, M. & Kuner, T. Three-Dimensional, Tomographic Super-Resolution Fluorescence Imaging of Serially Sectioned Thick Samples. *PLoS One* 7, e38098 (2012).
- Ehmann, N. et al. Quantitative super-resolution imaging of Bruchpilot distinguishes active zone states. *Nat. Commun.* 5, 4650 (2014).
- Körber, C. et al. Modulation of Presynaptic Release Probability by the Vertebrate-Specific Protein Mover. *Neuron* 87, 521–533 (2015).
- Dudok, B. et al. Cell-specific STORM super-resolution imaging reveals nanoscale organization of cannabinoid signaling. *Nat. Neurosci.* 18, 75–86 (2015).
- Spühler, I. A., Conley, G. M., Scheffold, F. & Sprecher, S. G. Super Resolution Imaging of Genetically Labeled Synapses in *Drosophila* Brain Tissue. *Front. Cell. Neurosci.* 10, (2016).
- Tang, A.-H. et al. A trans-synaptic nanocolumn aligns neurotransmitter release to receptors. *Nature* 536, 210–214 (2016).
- Heller, J. P., Odii, T., Zheng, K. & Rusakov, D. A. Imaging tripartite synapses using super-resolution microscopy. *Methods* (2019). doi:10.1016/j.ymeth.2019.05.024
- Berning, S., Willig, K. I., Steffens, H., Dibaj, P. & Hell, S. W. Nanoscopy in a Living Mouse Brain. *Science* (80-.). 335, 551–551 (2012).
- Kempf, C. et al. Tissue Multicolor STED Nanoscopy of Presynaptic Proteins in the Calyx of Held. *PLoS One* 8, e62893 (2013).
- Nishimune, H., Badawi, Y., Mori, S. & Shigemoto, K. Dual-color STED microscopy reveals a sandwich structure of Bassoon and Piccolo in active zones of adult and aged mice. *Sci. Rep.* 6, 27935 (2016).
- Chéreau, R., Saraceno, G. E., Angibaud, J., Cattaert, D. & Nägerl, U. V. Superresolution imaging reveals activity-dependent plasticity of axon morphology linked to changes in action potential conduction velocity. *Proc. Natl. Acad. Sci. U. S. A.* (2017). doi:10.1073/pnas.1607541114
- Tønnesen, J., Inavalli, V. V. G. K. & Nägerl, U. V. Super-Resolution Imaging of the Extracellular Space in Living Brain Tissue. *Cell* 172, 1108-1121.e15 (2018).

Table 2 for revision: Super-resolution microscope-based multiplex approaches.

Super-resolution method	Automation	Multiplex capacity*	Sample*	Quality	Tags	Tag removal	Publication
STORM							
3D dSTORM, spectral demixing	re-staining, microscopy	16	tissue: MNTB synapse	LP xy ~7 nm, LP z ~30 nm; RP xy ~17 nm	primary & secondary antibodies, primarily labeled antibodies and nanobodies, other labels	photobleaching: 5 min 405 nm & 661 nm lasers, elution: 0.1% SDS pH 13	Klevanski et al. (current not published paper)
2D STORM	n.a.	5	cells: BS-C-1	LP <10 nm; RP xy ~10 nm	primary & secondary antibodies	quenching: sodium borohydride	Tam et al. 2014
2D dSTORM	n.a.	4	cells: HeLa	SR xy >20 nm, SR z > 50 nm; RP xy ~10 nm	primarily labeled antibodies	photobleaching: 5 min 405 nm & 637 nm lasers, quenching: NaBH ₄	Valley et al. 2015
2D dSTORM	n.a.	25	cells: lymphocyte	LP xy ~2.6 nm; RP xy ~2 nm	primarily labeled antibodies	photobleaching: 647 nm laser (100 mW) ~2 s in 1× PBS; elution: 3.5 M MgCl ₂ pH 6, 20 mM PIPES, 0.1% Tween-2	Yi et al. 2016
PAINT							
2D & 3D DNA-PAINT	n.a.	10	DNA origami	SR xy <10 nm	origami structures, tubulin, peroxisomes, mitochondria, Golgi	washing in 5 mM Tris-HCl, 10 mM MgCl ₂ , 1 mM EDTA, 0.05% Tween 20, pH 8	Jungmann et al. 2014
Exchange-PAINT	n.a.	8	tissue: retina	LP xy ~5 nm, LP z =75 nm		washing with 0.1×PBS	Wang et al. 2017
3D DNA-PAINT & DNA-FISH	n.a.	3	cells: HeLa, mouse embryonic fibroblasts	SR xy ~20 nm, SR z ~80	PAINT: MacroH2A.1, Lamin B, nucleophosmin; FISH: major satellite, minor satellite, telomere regions	FISH: denatured for 3 min at 78 °C	Schueder et al. 2017

DNA-PAINT & dSTORM	re-staining, microscopy	5	cells: COS-7	SR xy ~68 to 97 nm	primary antibodies and DNA strand-conjugated secondary antibodies, phalloidin-Atto488	washing for PAINT	Almada et al. 2019
-------------------------	---	--------------	--------------------	---	-------------------	--------------------

MNTB = medial nucleus of the trapezoid body, (d)STORM = (direct) stochastic optical reconstruction microscopy, STED = stimulated emission depletion, PALM = photoactivated localization microscopy, PAINT = point accumulation for imaging in nanoscale topography, n.a. = no automation, SR = spatial resolution, LP = localization precision, RP = registration precision, *maximal number of targets reported

1. Tam, J., Cordier, G. A., Borbely, J. S., Sandoval Álvarez, Á. & Lakadamyali, M. Cross-Talk-Free Multi-Color STORM Imaging Using a Single Fluorophore. *PLoS One* 9, e101772 (2014).
2. Valley, C. C., Liu, S., Lidke, D. S. & Lidke, K. A. Sequential Superresolution Imaging of Multiple Targets Using a Single Fluorophore. *PLoS One* 10, e0123941 (2015).
3. Yi, J. et al. madSTORM: a superresolution technique for large-scale multiplexing at single-molecule accuracy. *Mol. Biol. Cell* 27, 3591–3600 (2016).
4. Jungmann, R. et al. Multiplexed 3D cellular super-resolution imaging with DNA-PAINT and Exchange-PAINT. *Nat. Methods* 11, 313–8 (2014).
5. Wang, Y. et al. Rapid Sequential in Situ Multiplexing with DNA Exchange Imaging in Neuronal Cells and Tissues. *Nano Lett.* 17, 6131–6139 (2017).
6. Schueder, F. et al. Multiplexed 3D super-resolution imaging of whole cells using spinning disk confocal microscopy and DNA-PAINT. *Nat. Commun.* 8, (2017).
7. Almada, P. et al. Automating multimodal microscopy with NanoJ-Fluidics. *Nat. Commun.* 10, 1223 (2019).

REVIEWERS' COMMENTS:

Reviewer #2 (Remarks to the Author):

I thank the authors for positively addressing in detail each of the points I have raised. I believe the manuscript is now at a stage of being ready for publication and I am happy to have had the opportunity to review what is an outstanding contribution to both the super-resolution microscopy and cell biology fields.

REVIEWERS' COMMENTS:

Reviewer #2 (Remarks to the Author):

I thank the authors for positively addressing in detail each of the points I have raised. I believe the manuscript is now at a stage of being ready for publication and I am happy to have had the opportunity to review what is an outstanding contribution to both the super-resolution microscopy and cell biology fields.

=> We thank the reviewer for the very positive assessment of our work and the thorough and constructive criticisms.